# Interrogating the viral dark matter of the rumen ecosystem with a global virome database

Ming Yan[1,2], Akbar Adjie Pratama [2,3], Sripoorna Somasundaram[1,2], Zongjun Li [4], Yu Jiang[4], Matthew B. Sullivan [2,3,5] & Zhongtang Yu [1,2]✉

The diverse rumen virome can modulate the rumen microbiome, but it remains largely unexplored. Here, we mine 975 published rumen metagenomes for viral sequences, create a global rumen virome database (RVD), and analyze the rumen virome for diversity, virus-host linkages, and potential roles in affecting rumen functions. Containing 397,180 species-level viral operational taxonomic units (vOTUs), RVD substantially increases the detection rate of rumen viruses from metagenomes compared with IMG/VR V3. Most of the classified vOTUs belong to *Caudovirales*, differing from those found in the human gut. The rumen virome is predicted to infect the core rumen microbiome, including fiber degraders and methanogens, carries diverse auxiliary metabolic genes, and thus likely impacts the rumen ecosystem in both a top-down and a bottom-up manner. RVD and the findings provide useful resources and a baseline framework for future research to investigate how viruses may impact the rumen ecosystem and digestive physiology.

A flurry of recent virus-focused metagenomic studies have generated very large viral genome catalogs and databases for several ecosystems, including ocean viruses[1,2], the human gut[3–5], and soil[6]. They revealed vastly diverse viromes, identified numerous auxiliary metabolic genes, and shined new light on the ecological impact of viruses. Furthermore, model system-focused studies have begun to reveal how viruses can reprogram the metabolism of their prokaryotic hosts by forming distinct virocells that alter the ecological fitness and metabolism of the hosts[7]. Emerging evidence supports the potential impacts of viruses on ocean biogeochemistry[1,8], human physiology[4], and disease states[9]. No similar studies on the rumen virome or rumen-specific virome database are available.

The rumen harbors a diverse multi-kingdom ecosystem containing bacteria, archaea, fungi, protozoa, and viruses. Collectively, the rumen microbiome digests and ferments otherwise indigestible feedstuffs and provides most of the energy (in the form of volatile fatty acids) and metabolizable nitrogen (in the form of microbial protein) needed by ruminants to grow and produce meat and milk. Strong associations of rumen bacteria, archaea, and protozoa with feed efficiency, methane ($CH_4$) emissions, and animal health have been documented[10], but rumen viruses, abundant notwithstanding, remain poorly understood, despite virus-focused studies contributing to the characterization of the rumen virome[11,12]. Early studies using electron microscopy documented morphologically diverse bacteriophages and revealed the predominance of tailed phages[13,14]. Early culture-dependent studies found bacteriophages that could infect a broad range of species or strains of rumen bacteria, including prevalent species of *Prevotella*, *Ruminococcus*, and *Streptococcus*, and classified most of these phages based on their morphology into the families *Myoviridae*, *Siphoviridae*, *Podoviridae*, and *Inoviridae* (reviewed by Gilbert and Klieve[15]). Although these studies provided valuable information on rumen viruses, the simple morphologies of phages do not allow reliable taxonomic classification, and thus, the International Committee on Taxonomy of Viruses (ICTV: https://ictv.global/taxonomy) no longer recognizes morphology-based virus classification.

[1]Department of Animal Sciences, The Ohio State University, Columbus, OH, USA. [2]Center of Microbiome Science, The Ohio State University, Columbus, OH, USA. [3]Department of Microbiology, The Ohio State University, Columbus, OH, USA. [4]College of Animal Science and Technology, Northwest A&F University, Yangling, China. [5]Department of Civil, Environmental, and Geodetic Engineering, The Ohio State University, Columbus, OH, USA. ✉e-mail: yu.226@osu.edu

Genomics, metagenomics, and metatranscriptomics have become the primary technologies for studying viromes, including the rumen virome. Recent culture-dependent whole-genome sequencing identified 10 phages that infect *Prevotella ruminicola*, *Ruminococcus albus*, *Streptococcus bovis*, and *Butyrivibrio fibrisolvens*[16,17], which play important roles in feed digestion and fermentation. These phage genomes display modular genomic organization, conserved viral genes, and the potential to be both lytic and lysogenic[17]. Rumen viruses have also been studied using metagenomes of virus-like particles (VLPs) (reviewed in[11]). However, the reference genome databases that have been used underrepresent rumen viruses, thus limiting the identification and classification of rumen viruses and the prediction of their host. For example, rumen viruses with diverse genotypes have been found, but most of them have not been classified due to a lack of matches to reference viral sequences[18–20]. Miller et al.[18] found clustered regularly interspaced short palindromic repeat (CRISPR)/CRISPR-associated protein (Cas) elements in some rumen microbial genomes and metagenomes but found few spacer sequences matching rumen viral sequences for host prediction. Therefore, it has been difficult to characterize the rumen virome, especially concerning novel viruses.

Recent bioinformatics tools specific for viral analysis (e.g., CheckV[21], VirSorter2[22], and VIBRANT[23]) and increasing genomic resources (e.g., efam[24]) greatly facilitate viral identification from metagenomic sequences, and sequence-space organizational strategies provide scalable viral classification[25] and taxonomy[26]. These tools enabled the development of large niche-specific virome databases[1,5] and comprehensive global virome studies[27]. Using metagenomic sequencing and the above bioinformatics tools, two recent studies identified diverse rumen viruses and explored their nutritional implications[28,29]. These two studies, with small sample sizes (5 beef steers[29] or one moose[28]), identified approximately 2000 viral populations (unique contigs) of a diverse rumen virome and its potential importance to the rumen ecosystem. Not surprisingly, given the nascent nature of the field, the ability to predict hosts of those viral populations was low (hosts predicted only for 3 viral populations at the phylum level[29] and 113 viral populations at the MAG level[28]). Motivated by comprehensive virome studies[3–5] and leveraging the many publicly available rumen metagenomic datasets, we aimed to analyze the global rumen virome by analyzing 20 TB of sequence data from nearly 1000 metagenomic datasets from diverse ruminants, both domesticated and wild, across 5 continents. We developed a systematically cataloged rumen virome database (RVD) that contains nearly 400,000 species-level viral operational taxonomic units (vOTUs), explored the core rumen virome, predicted bacteria, archaea, and protozoa that the identified viruses are likely to infect, and inferred the potential ecological roles of the rumen viruses from auxiliary metabolic genes (AMGs) and antimicrobial resistance genes (ARGs) carried by the rumen viral genomes. We also tested RVD to determine if it could improve the identification of viruses from metagenomic sequences. By expanding the diversity of rumen viruses recorded in NCBI RefSeq Viral (by >12-fold) and IMG/VR V3 and improving the identification of viral sequences based on rumen metagenomics, RVD will be useful as a new community resource and will provide new insights for future studies on the rumen virome and its implication in feed digestion, microbial protein synthesis, feed efficiency, and $CH_4$ emissions.

## Results

### Rumen viruses are highly diverse and represent unique lineages

Using state-of-the-art bioinformatics tools (Supplementary Fig. 1), we characterized the global rumen virome by analyzing 20 TB of sequences derived from 975 rumen metagenomes (Supplementary Data 1) that were sampled from 13 ruminant species or different husbandry regimes across 8 countries on 5 continents (Fig. 1a, b). Following the recommendations of a recent viromics benchmarking paper[30] and stringent criteria, we identified 705,380 putative viral contigs of >5 kb each and clustered them into 411,125 vOTUs. After validation with VIBRANT[23], we constructed a rumen virome database (RVD, download available at https://zenodo.org/record/7412085#.ZDsE2XbMK5c) representing 397,180 vOTUs (Supplementary Fig. 1), with 193,327 vOTUs of >10 kb. Checking with CheckV[21] revealed 4400 complete vOTUs, 4396 high-quality vOTUs, and 32,942 medium-quality vOTUs. The completeness and quality of the RVD vOTUs were probably underestimated because CheckV is database dependent, and the databases used are primarily derived from other ecosystems. All the vOTUs in RVD meet Uncultivated Virus Genome (MIUViG) standards[25].

We were able to classify 1,857 vOTUs (0.47% of the total) as belonging to existing genera and 32,934 vOTUs (8.3%) to existing families (Supplementary Data 2). Most of the genus- and family-assignable vOTUs (98.4%) were assigned to *Siphoviridae*, *Myoviridae*, or *Podoviridae* in the order *Caudovirales* (Fig. 2a, b), which are also the most abundant families in the human gut virome[5]. Compared with a phylogenetic tree (Fig. 2c), the human and rumen viromes shared only 14% of the genus-level taxa of *Caudovirales* (Fig. 2d), reflecting the divergence between the two types of viromes. The remaining vOTUs (91.7% of the total) could not be assigned to any existing genera or families, and they thus represent new lineages. Additionally, 121 vOTUs (0.03%) were identified as crAss-like viruses. RVD also contains protozoan viruses (or endogenous viral elements, EVEs), some of which were assigned to *Phycodnaviridae*, *Mimiviridae*, and *Retroviridae*. Some of the rumen viruses were predicted to infect rumen archaea (thus archaeaphages). Although the metagenomes primarily consisted of dsDNA, we also identified 109 vOTUs as ssDNA viruses. All were assigned to the family *Microviridae*. Rarefaction analysis (Fig. 2e) revealed that more rumen viruses remain to be identified.

### RVD facilitates the analysis of the rumen virome from metagenomic sequences

We tested RVD for its utility in rumen virome analysis with five sets of metagenomic sequences. First, we compared RVD with the IMG/VR V3 and NCBI RefSeq Viral databases using two recent independent sets of rumen metagenomes that were not included in any of the three databases. Substantially more viral sequences were identified in RVD than in IMG/VR V3 (Supplementary Fig. 2a), and none of the metagenomic sequences could be mapped to NCBI RefSeq Viral. Then, we tested the estimation of viral abundance in RVD. We found no significant difference among the five ruminant species (Supplementary Fig. 2b), but significantly higher viral abundance was found in the rumen metagenomes of both goats and dairy cows suffering from subacute rumen acidosis (SARA, a common metabolic disorder in ruminants fed starch-rich diets) (Supplementary Fig. 2c, d). Using RVD, we also reanalyzed recent metagenomic sequences derived from 5 beef steers[29] and one moose[28]. From the steer rumen metagenomes, we identified 706 vOTUs, including 4 archaeaphages and achieved genus-level taxonomic classification, while the authors[29] could only assign their vial contigs to viral families, and they could not identify any archaeaphages. We predicted the hosts down to the species level for 113 of the vOTUs, while the authors could only predict hosts at the phylum level for 3 of their vial contigs. Similarly, using RVD, we identified 789 vOTUs from the moose metagenome[28] and were able to predict hosts down to the species level for 126 of the vOTUs, while in the original paper, hosts were predicted only at the phylum level for 113 of the viral contigs. These results indicate that RVD can significantly improve the detection, identification, and taxonomic assignments of rumen viruses from metagenomic sequences and can better predict virus–host linkages.

We developed RVD with sequences of bulk metagenomes, although a few studies have used sequences derived from VLP-enriched metagenomes. We thus evaluated RVD for the analysis of

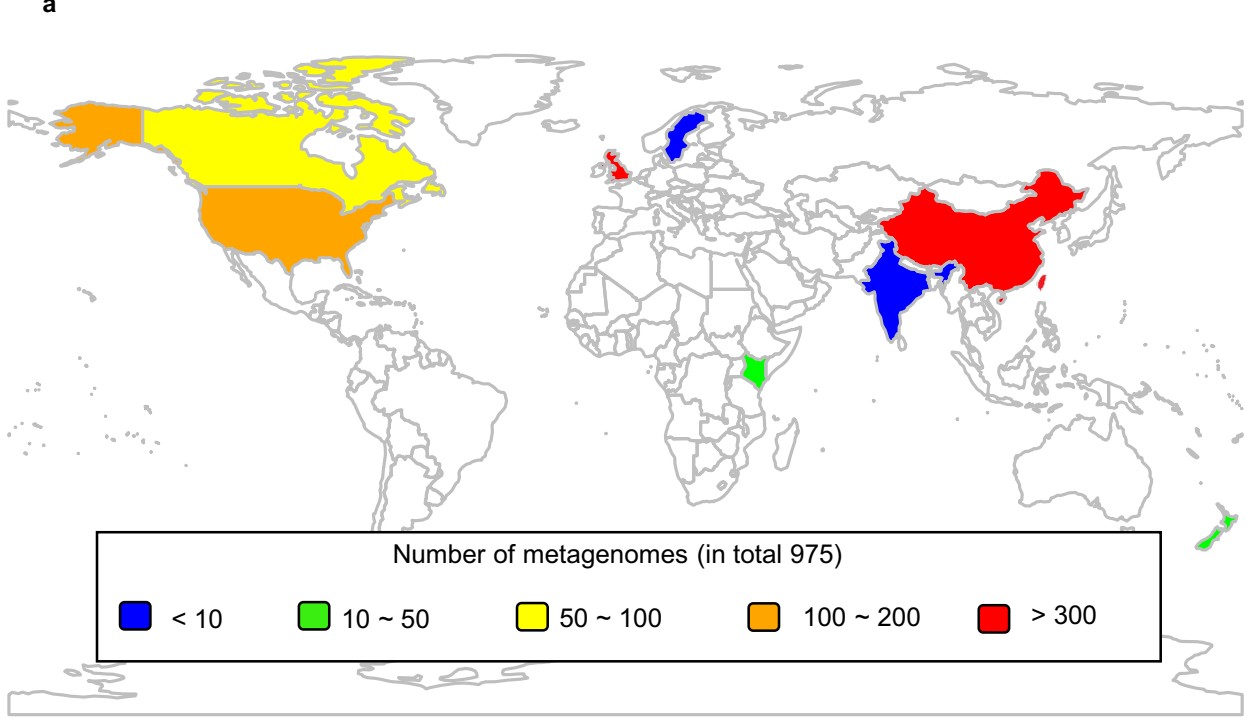

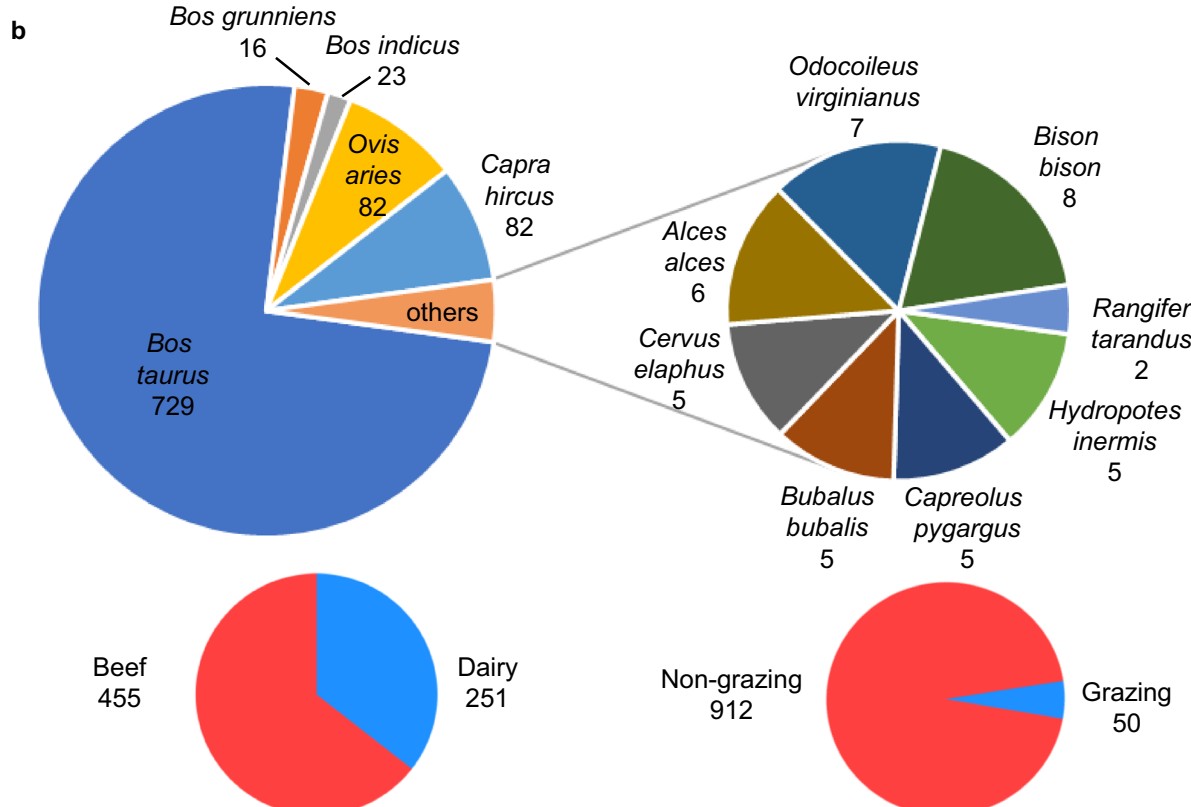

**Fig. 1 | Rumen viral genomes recovered from 13 ruminant species or animal husbandry regimes across 5 continents. a** A global heatmap showing the geographic distribution of the 975 rumen metagenomes. **b** Number of rumen metagenomes from different ruminant species or production husbandries.

rumen viromes in bulk metagenomes vs. VLP-enriched metagenomes. Using the two types of rumen metagenomes derived from the same 5 steers[29], we found that, as expected, VLP-enriched metagenomes contained a higher proportion of viral sequences (Supplementary Fig. 3a), but the two types of rumen metagenomes showed comparable percentages of lytic viruses (Supplementary Fig. 3b). However, the bulk metagenomes presented a higher percentage of vOTUs that were represented in RVD (Supplementary Fig. 3c). Hence, VLP-enriched metagenomes are needed to expand RVD with viruses that are typically underrepresented in bulk metagenomes.

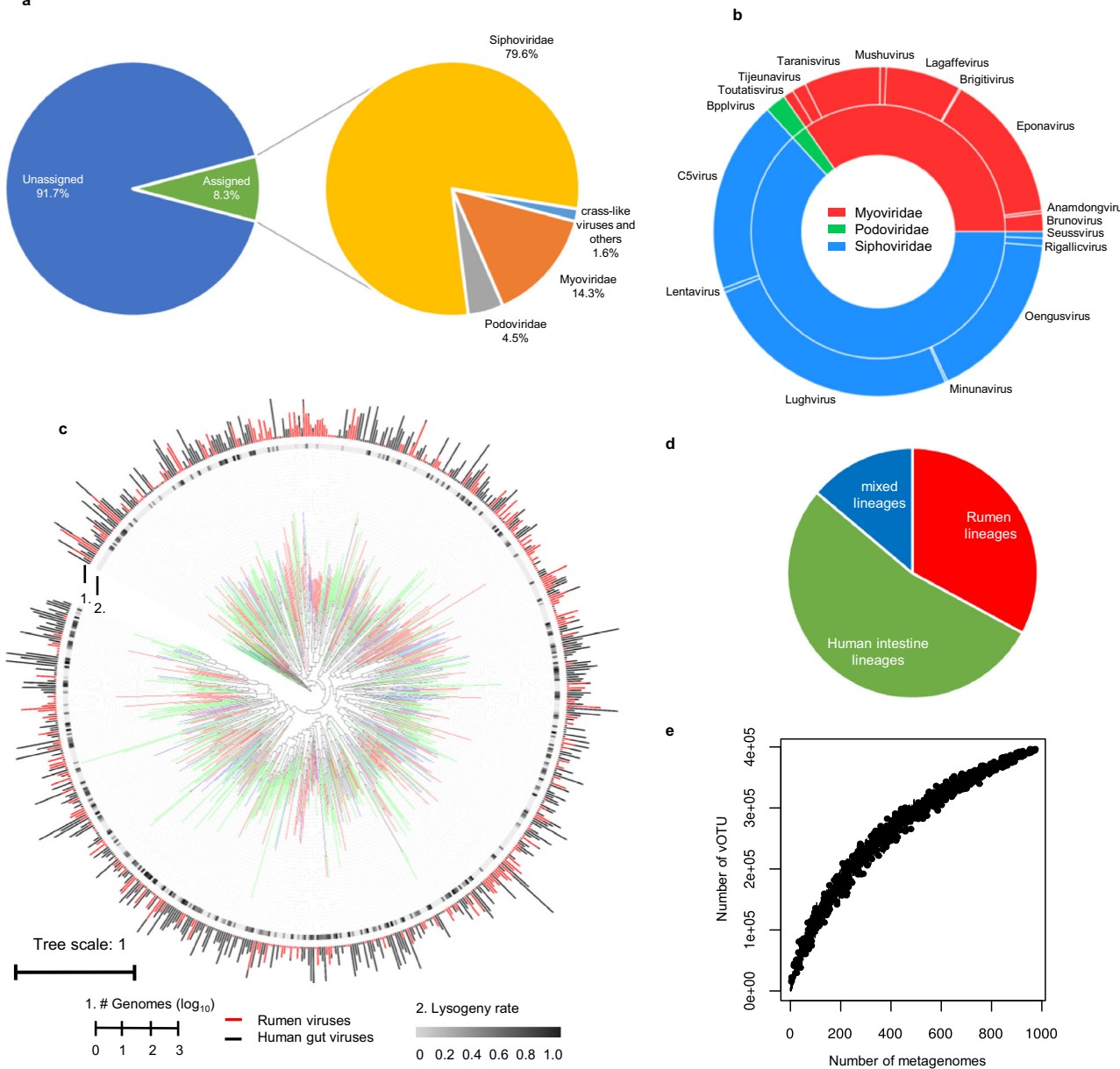

**Fig. 2 | Taxonomic classifications of the rumen viruses. a** Family level taxonomy and proportion of the rumen viruses in the rumen virome database RVD). Most of the vOTUs (99.7%) classified to existing genera or families were under the order of *Caudovirales*, including 121 identified crAss-like phages. Detailed taxonomic assignments of individual vOTUs are presented in Supplementary Data 2. **b** Genus-level taxonomy and proportion of the 1,858 vOTUs that could be assigned to existing genera or families using vConTACT2. **c** A phylogenetic tree of *Caudovirales* viruses built with 77 concatenated marker genes identified in 10,203 vOTUs with a >50% completeness of the current study and the two largest human gut virome databases (MGV[4] and GPD[5]). For better visualization, only one representative vOTU (the longest and most complete) was included for each genus-level vOTU (714 in

total). The branches were color-coded: green, the *Caudovirales* lineages exclusively found in the human virome; red, the lineages exclusively found in the rumen virome of the current study; blue, the lineages found in both the rumen and the human viromes. Lysogeny rates (proportion) were calculated with VIBRANT and shown as the inner ring. The number of vOTUs representing each lineage was shown as a bar plot (red for human viruses, and black for human viruses). **d** Proportion of lineages of *Caudovirales* viruses unique to the human intestine, the rumen, and shared. **e** A rarefaction curve of the vOTUs identified in the rumen virome. The upward trend of the rarefaction curve indicates that more rumen viruses remain to be identified at the specie level.

## Rumen viruses have a broader range of hosts than human gut viruses

Host prediction is important to understand the potential roles of viruses in an ecosystem. By predicting the probable hosts of the identified rumen viruses, we identified 2403 archaeophages and 40,881 bacteriophages. The archaeaphages were predicted to infect 25 genera of archaea, and the bacteriophages could infect 1051 genera of bacteria, including genera with well-characterized species such as *Methanobrevibacter* (e.g., *M. ruminantium, M. millerae*), *Fibrobacter*

(e.g., *F. succinogenes*), *Prevotella* (e.g., *P. bryantii, B. ruminicola, P. multiformis*), *Ruminococcus* (e.g., *R. albus, R. callidus, R. flavefaciens*), and *Streptococcus* (e.g., *S. equinus*) (Supplementary Data 3). Surprisingly, a high proportion of bacteriophages (9214, or 22.5%) and archaeophages (396, or 16.5%) were predicted to infect multiple host species. In addition, 3.8% (1544) of these broad-host-range rumen phages can infect species across multiple bacterial phyla (Fig. 3a). In comparison, only 0.13% of human gut phages have a broad host range[3]. The cross-phylum host range of rumen phages suggests their potential

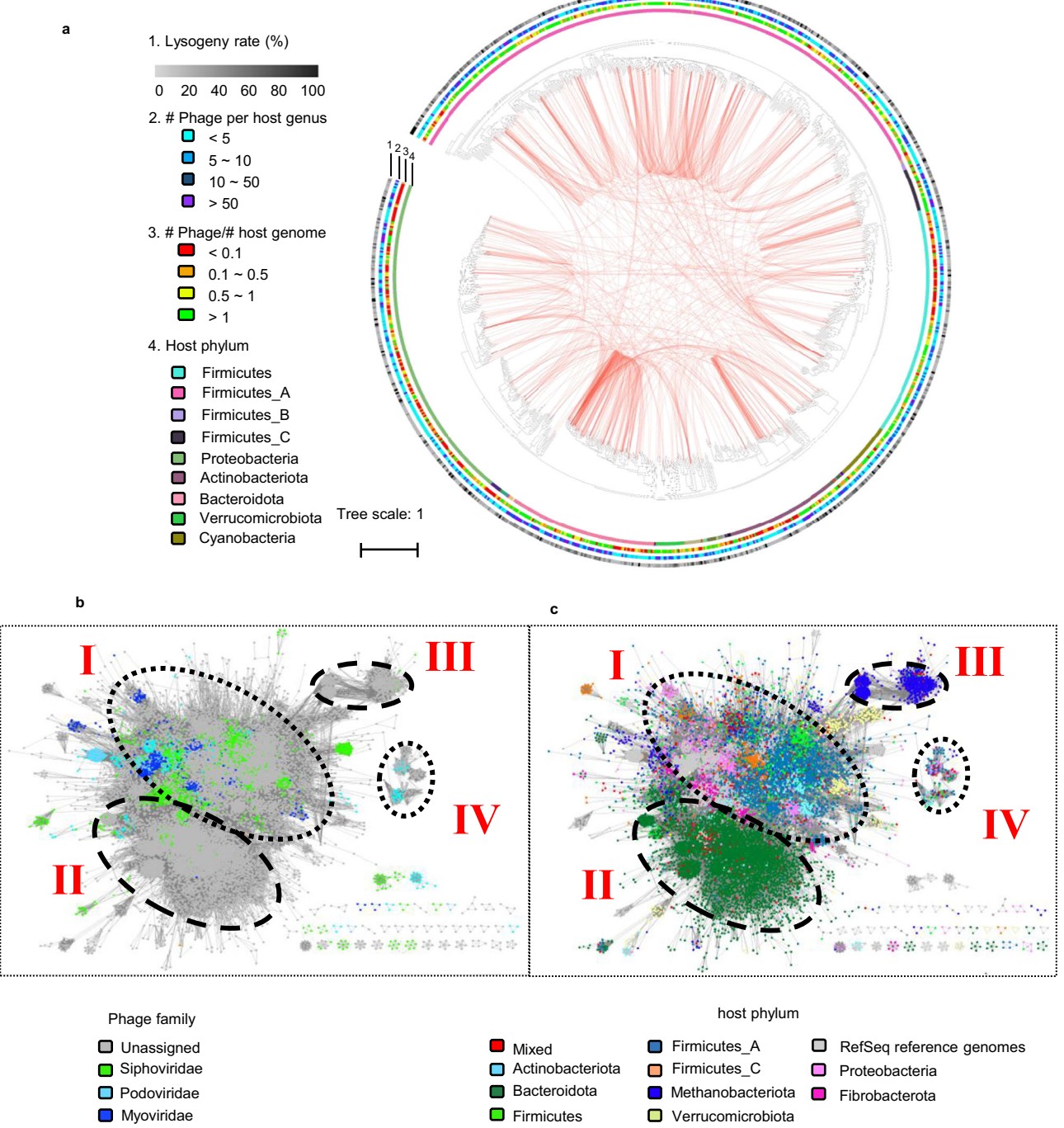

**Fig. 3 | Bacterial host range of the rumen viruses. a** A genome-based phylogenetic tree of 1051 bacterial genera that contained the predicted hosts of 40,881 vOTUs. The hosts were inferred by (i) aligning the sequences of the representative vOTUs (the longest with the highest completeness) of each vOTUs with 22,087 metagenome-assembled genomes (MAGs) of rumen bacteria and 242,387 bacterial reference genomes of NCBI RefSeq and (ii) aligning the CRISPR spacer sequences of the representative vOTUs and those of the prokaryotic genomes and MAGs. The prokaryotic genomes and MAGs were classified using GTDB-Tk. The phylogenetic tree was constructed with the genomes or MAGs of the inferred hosts (clustered into genera) and their predicted phages to examine the lysogeny rate (%), the number of phages per host genus, and the number of phages per host genome/

MAG. In total, 4394 vOTUs likely have a host range across multiple genera. These genera are connected with orange arcs. The rings correspond to lysogeny rates (ring 1, calculated based on VIBRANT results), number of phages per genus of hosts (ring 2), number of phages per host genome/MAG (ring 3), and bacterial phyla to which the bacterial host genera belong (ring 4). **b, c** Gene-sharing network of the vOTU with predicted host obtained from vConTACT2 and visualized using Cytoscape, with the viral genomes being colored according to the family assignment (**b**) and predicted hosts colored according to phyla (**c**). See also Supplementary Figs. 1 and 2 for the host assignment of archaea and protozoa and Supplementary Data 3 for the detailed host assignments of individual vOTU.

in mediating genetic exchange across phylum boundaries, which can facilitate microbial adaptation and evolution[3,31]. Thirty-eight of the 52 single-cell amplified genomes (SAGs) of rumen ciliates, which represented 19 ciliate species across 13 genera, had predicted EVEs.

We calculated the lysogenic rate (% of the total), the average number of phages per host genus, and the average number of phages per individual host genome for bacteriophages (Fig. 3a) and archaeophages (Supplementary Fig. 4). The number of phages per host genome varied, with most of the hosts belonging to *Proteobacteria* showing <0.1 phage per host genome, whereas most of the hosts of *Firmicutes* A have >1 phage per host genome. The percentage of lysogenic viruses varied among the host genera, and it was low for most host genera (Fig. 3c). Most ciliate SAGs presented multiple EVEs, among which all five SAGs of *Isotricha* sp. YL-2021b and *Dasytricha ruminantium* presented the greatest number (>50) EVEs per SAG (Supplementary Fig. 5). Little is known about viruses infecting ciliates, and no EVEs have been reported for even model ciliate species (e.g., *Tetrahymena thermophila*). However, EVEs have been recently found in *Entamoeba* and *Giardia* in human stool metagenomes[32]. Therefore, rumen ciliates probably carry EVEs. The large number of EVEs per ciliate SAG may correspond to the high polyploidy and the enormous numbers of chromosomes found in many rumen ciliates (e.g., >10,000 in *Entodinium caudatum*[33]).

Sequence similarity analysis revealed that homologous phages usually infect homologous human gut MAGs[31]. However, such analysis cannot classify viruses with variable evolutionary modes and tempos[26]. To better predict virus–host linkages and gene flows within these relationships, we generated a gene-sharing network of all the phages with a host match and visualized the network based on their taxonomy (Fig. 3b) and their host phyla (Fig. 3c). The 43,284 vOTUs with a host match formed 2,764 genus-level clusters, but only 218 of the clusters included one or more viral genomes from the NCBI RefSeq Viral database. Thus, RVD greatly improved virus–host linkage analysis at the genus level compared to NCBI RefSeq Viral (by >12-fold). Based on the gene-sharing network, most rumen vOTUs were clustered into four groups (Fig. 3b). Groups I (the largest) and IV (the smallest) contained more classified vOTUs than groups II and III. Groups I and IV had a broader host range among bacterial phyla, including both gram-positive and gram-negative bacteria with different niches and capacities, but few of their genera or families were predominant in the rumen. Groups II and III mainly infected *Bacteroidota* and *Methanobacteriota*, respectively (Fig. 3c), and most viruses of these two groups could not be classified with any of the current virome databases; thus, they represent new viral lineages. The narrow host range (a single phylum) of groups II and III supports the notion that phages with a high degree of gene sharing generally infect phylogenetically related hosts.

## Rumen virus AMGs potentially alter the metabolism and ecological fitness of their hosts

By searching only the complete viral contigs (vMAGs, 5912 in total), we found 286 vMAGs carrying more than one AMG (see Supplementary Data 4 for detailed annotation and curation results). These AMGs represented 41 distinct categories, including 36 identified in previous studies and 5 that had not been identified in other viromes (Fig. 4a). The 5 new categories of AMGs were each carried by more than two vMAGs. They encode enzymes involved in a wide range of metabolic processes, including the metabolism of nucleotides, carbohydrates, vitamins, and nitrogen.

Notably, the most prevalent AMG encoded DNMT1, a DNA (cytosine-5)-methyltransferase that protects viruses from the antiviral restriction-modification systems of their hosts[34,35], which was found in more than 100 of the vMAGs. This concurs with the high prevalence of this type of AMG found in the ocean virome[36]. Such an AMG probably represents a counter-defense mechanism of rumen viruses. In addition to directly augmenting host nucleotide metabolism to benefit viral

replication, some viruses use AMGs to facilitate nutrient acquisition by their host to indirectly improve viral survival. Interestingly, all 5 new AMGs found in the rumen virome were related to nutrient acquisition and biosynthesis. In particular, several AMGs found in the vMAGs encoded glycoside hydrolases (GHs), including GH16 and G114, which are important enzymes involved in feed fiber digestion, and asparagine synthase (Fig. 4a), which mediates ammonia assimilation into asparagine. In addition, we found AMGs related to vitamin B12 (i.e., cobaltochelatase CobS gene) and the synthesis of aromatic amino acids (i.e., chorismate mutase gene). We found no cellulase-encoding AMGs among the vMAGs, but two AMGs encoding cellulases were found among the incomplete viral contigs: one GH5 and one GH9, each of which was flanked by viral genes on both sides (Fig. 4b). This GH9 shared 79.5% amino acid identity with a GH9 of a rumen MAG (GenBank: MBR3901505.1, classified as uncultured *Ruminococcus*), while the GH5 was 45.9% identical to a GH5 in another rumen MAG (GenBank: MBR3315497.1, classified as uncultured *Atopobiaceae*). The cloning and expression of both AMGs in *E. coli* showed that they encode functional cellulases hydrolyzing carboxymethylcellulose (Fig. 4c, d). By providing diverse and unique AMGs, rumen viruses may augment nutrient acquisition by their hosts or reprogram important host metabolic pathways.

## Rumen viruses carry a few types of antibiotic resistance genes (ARGs) but may facilitate ARGs transmission across phylogenetic boundaries

Among the 705,380 total viral contigs, 24 were found to carry ARGs, including several viral contigs carrying ARGs flanked by at least two viral structural genes (Fig. 5b and Supplementary Data 5). The dearth of viral ARGs corroborates the sparsity of ARGs among phage genomes[37]. However, we found several major ARG classes, including *tet*(W) and *tet*(O) (Fig. 5a and Supplementary Fig. 6), both of which are prevalent and highly expressed in rumen microbiomes[38–40]. Three ARG-carrying viral contigs recovered from three animals of the same herd were shown to carry the same ARGs (*van*(G) and van(W-G)) and presented nearly identical genomic architectures (Fig. 5b), pointing to the potential of rumen viruses as a route of ARG transmission between animals. However, given the limited number of ARG-encoding phages identified, the lack of on-farm antibiotics usage data, and the possibility that the ARGs identified might be present in the rumen ecosystem even if the respective antibiotics are not used, future ecological studies are needed to further explore the potential role of viruses as a reservoir of ARGs.

## The rumen virome is highly individualized, but a "core" virome exists across cattle receiving different dietary concentrate (grain) levels

We compared the prevalence of the individual vOTUs that were derived from the rumen metagenomes of 283 cattle fed three levels (low, medium, and high) of dietary concentrate to determine if a core rumen virome could be found. A core rumen virome was defined as vOTUs found in at least 50% of the animals fed each level of dietary concentrate. We found that most of the vOTUs were found in only one animal or multiple animals fed the same level of concentrate (Fig. 6a), indicating highly individualized rumen viromes among these cattle. The high between-animal variation was consistent with previous results[19]. A core rumen virome (approximately 1% of all vOTUs) was found for each concentrate level. Only 14 vOTUs, or 1.4% of the core vOTUs associated with the individual concentrate levels, were found to be core vOTUs across all three concentrate levels (Fig. 6b). In another set of rumen metagenomic data from animals of the same species fed diets with different concentrate levels and one set of rumen metagenomic data from different breeds of animals, we also found fewer vOTUs shared among animals across concentrate levels or across breeds compared to those within the same concentrate level or within

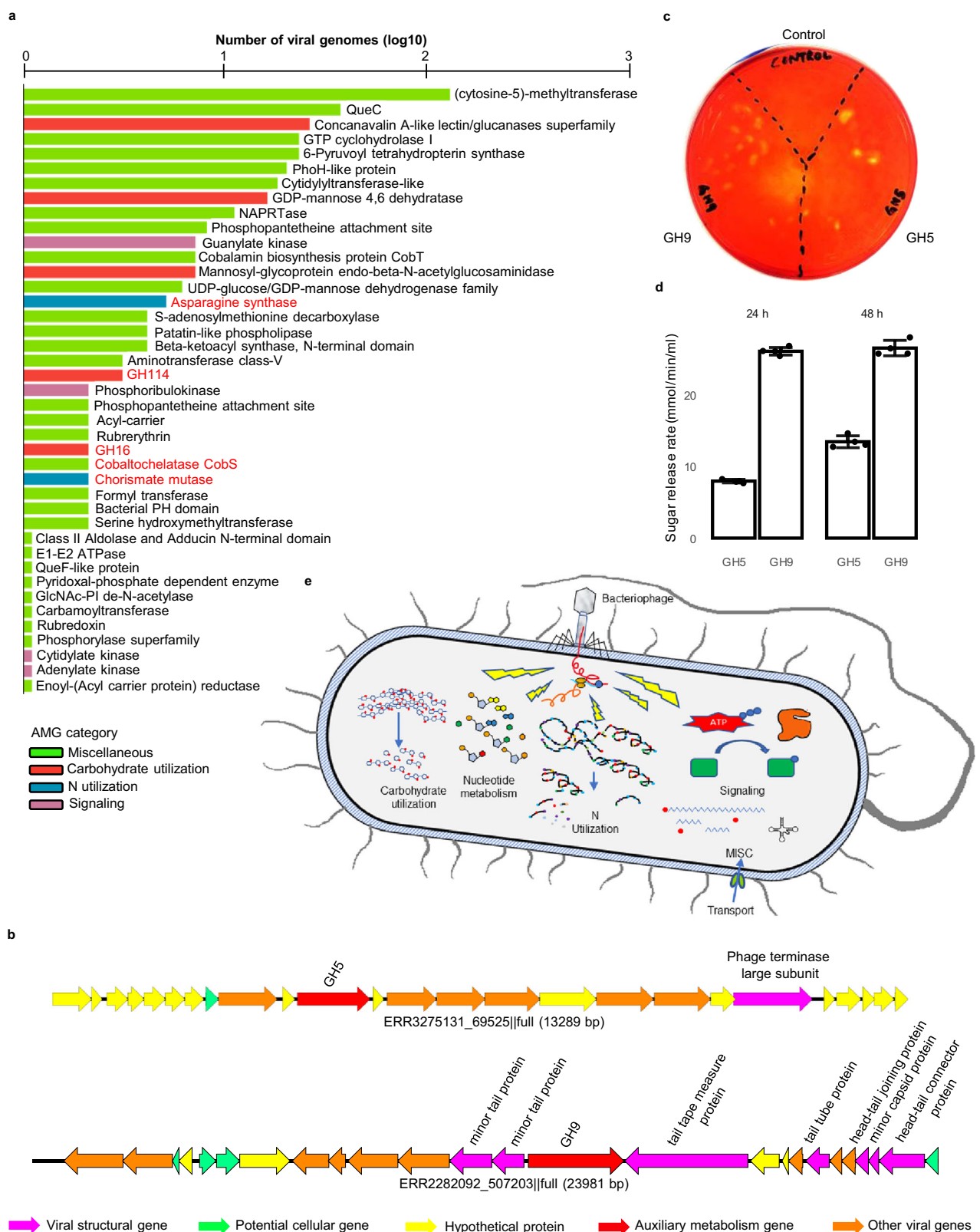

the same breed, respectively (Fig. 6c, d). The core rumen virome was predicted to mostly infect species of the core rumen bacteria, especially species of *Prevotella* (Supplementary Data 6). Thus, the core rumen virome was linked to the core rumen microbiome. We identified 121 crAss-like vOTUs (Supplementary Data 2), but unlike what was found in the human gut virome, none of them was included in the rumen core virome. The small core rumen viromes mirrored the highly

individualized rumen virome, which was consistent with early rumen virome studies even though they used different types of analyses (pulsed-field gel electrophoresis[41] or sequencing of VLP-enriched metagenomes[29]).

We further examined whether viral alpha- or beta-diversity differed among animal species, geographic locations, or studies (research projects), and we observed significant differences among all of these

**Fig. 4 | Auxiliary metabolic genes carried by rumen viruses. a** A bar plot showing the categories of AMGs and their occurrence ($\log_{10}$) identified in the rumen virome. Only the AMGs that had also been identified in previous studies and the AMGs that are identified in more than two vMAGs of the current study (highlighted in red) were shown. AMGs were predicted from vMAGs (286 in total) that passed a series of curation. See Supplementary Data 4 for the detailed AMGs curation process, full annotation of the final AMG-carrying vMAGs, and AMG functional category annotation. **b** The genomic organization of two viral contigs that carry one cellulase (GH5 or GH9) flanked by two viral genes at both sides. **c** A carboxymethylcellulose agar plate showing *E. coli* colonies that carry cloned viral GH5, GH9, or the cloning vector without any exogenous gene. The yellow halos indicate digestion of CMC after staining with Congo red. **d** Sugar release rate of *E. coli* broth cultures containing carboxymethylcellulose after incubation at 37 °C for 24 and 48 h ($n = 4$). The error bars denote the standard error. The center of the error bar represents the average of the replicates. **e** A conceptual model illustrating how AMGs might enable rumen viruses to modulate certain host metabolism.

categories (Supplementary Fig. 7). Permutational multivariate analysis of variance (PERMANOVA) showed that a much larger proportion of the variance was explained by the studies than by animal species or geographic locations. Thus, differences among animal species and geographic locations should be interpreted with caution because the differences could be confounded by variations in diets and feeding regimens among studies. We further examined what might have driven the "study effect" via hierarchical clustering of the studies based on shared vOTUs. The studies were hierarchically clustered into five groups, but none of the groups corresponded to an animal species, geography, method of rumen sample collection, or husbandry regime (Supplementary Fig. 8). Therefore, the observed variations in rumen viromes could likely be attributed to the multiple factors mentioned above, especially diets, which have profound effects on the rumen microbiome.

## Discussion

The vast diversity and potential ecological impact of viruses in the environment[1,42] and the human gut[3,4] are becoming increasingly well documented and recognized. Although the viruses in the rumen microbiome are also thought to be diverse and can impact the rumen microbiome and the major rumen functions (e.g., feed digestion, fermentation, microbial protein synthesis, methane emissions), they are much less well characterized and understood[10,11]. By mining most of the published rumen metagenomes (nearly 1000), we recovered nearly 400,000 vOTUs and created a global rumen virome database, RVD. This database greatly expands upon the current catalogs of rumen viruses and shines new light on their diversity and potential impacts across major ruminant species and animal husbandry regimes. Most of the detected rumen viruses represent new viral lineages, as ~91% could not be classified into any existing viral species, genera, or families, which concurs with previous viral studies in beef steers[29] and moose[28].

As a database specific to the rumen ecosystem, RVD greatly facilitates and improves the detection, identification, and taxonomic assignments of rumen viruses from metagenomic sequences and better predicts virus–host linkages. It will be a useful resource in future rumen virome studies. However, RVD is far from complete, as it still lacks or underrepresents some of the rumen viruses, as shown by the rarefaction analysis. Several types of viruses are missing in particular. First, viruses with smaller genomes (<5 kb), including ssDNA viruses, should be added. Since ssDNA viruses are enriched in VLP metagenomes due to size filtration and DNA amplification[5], viral metagenomics combined with a smaller viral genome size cutoff will help capture rumen viruses with a small genome size and ssDNA genome in future studies. Second, RNA viruses have RNA genomes, so unless they are endogenous virus elements, they are rarely detected by metagenomics or viral metagenomics. As demonstrated in oceans[8,42] and soils[43], RNA viruses are also likely diverse and important in the rumen ecosystem. Indeed, the only rumen RNA virus study using metatranscriptomics obtained 2466 unique RNA viral contigs, which were assigned to nine viral families[44]. Comprehensive analysis of rumen metatranscriptomes, both published and future, will allow the identification of rumen RNA viruses. Third, because only a few genomes of rumen fungi are available, we did not analyze the mycoviruses in the RVD with respect to identification and host prediction. As more genomes of rumen fungi

become available in the future, the mycoviruses in RVD can be identified, and their probable hosts can be predicted. Although the database needs to be improved, RVD is a useful resource for analyzing rumen viromes in bulk metagenomes and VLP-enriched metagenomes, including both previously published and new datasets. Furthermore, these data are critical for contextualizing new sequences for viral classification, analyses of alpha- and beta-diversity and abundance, and predictions of virus–host linkages.

In addition to detection, identification, and classification, it is essential to predict hosts down to the species or genus level to understand the ecological roles of viruses in any ecosystem, including the rumen ecosystem. We achieved host prediction for many archaeophages (>2400) and bacteriophages (>40,000) down to the species level, and many of the host species are known to play important roles in feed digestion, fermentation, and methane emissions. Advancement in the prediction of hosts and virus–host linkages will aid in understanding the ecological roles of rumen viruses. Such information will be especially useful when both the rumen metagenome and virome are investigated for their association with major rumen functions. Among the rumen vOTUs with a predicted host match, 99.5% were inferred to infect prokaryotes primarily found in the rumen, even though most of the reference prokaryote genomes that were used came from prokaryotes in other environments, demonstrating the rigor and low false positive rate of our host prediction pipeline.

Corroborating previous studies on viromes in the human gut[3,9,31], we found rumen viruses that likely infect multiple species, even species in different phyla. Indeed, a combination of long-read assembly and proximity ligation verified that a rumen virus could infect 11 distinct bacterial genera[40]. Interestingly, compared to the viruses in the human gut, more rumen viruses could have a broader host range, as evidenced by the high proportion of rumen vOTUs predicted to infect multiple host species across phylogenetic boundaries, even phylum boundaries. This discrepancy may stem from the more diverse microbiome and physiological conditions in the rumen (due to large variations in animal species/genetics and diets) than in the human gut. Indeed, all 32 phyla of rumen bacteria that are represented by the reference genomes could be infected by phages, while only 8 phyla were likely to be hosts of human gut phages. Phages mediate horizontal gene transfer through transduction, both general and specialized, thereby diversifying the gene repertoires of the host, which leads to strain-level heterogenerity[45] and microbial adaptation and evolution[3]. The numerous prophages predicted to infect diverse hosts across phylogenetic boundaries point to their potential to facilitate host evolution. However, as in other studies, the hosts of rumen viruses were predicted solely based on DNA sequence similarity. Experimental validation is needed to verify the true hosts of particular rumen viruses of interest, and the host range predicted in the current study will be helpful for such studies.

The rumen ecosystem has a core rumen microbiome that contains species of predominant genera of bacteria (e.g., *Prevotella* and *Ruminococcus*), archaea (e.g., *Methanobrevibacter*), and protozoa (e.g., *Entodinium*) and plays a major role in major rumen functions. To determine if a core rumen virome also exists, we examined the virome profiles across all rumen microbiomes. We did not find a "global core" rumen virome and instead identified a highly individualized rumen virome. This probably occurred because the large number of rumen

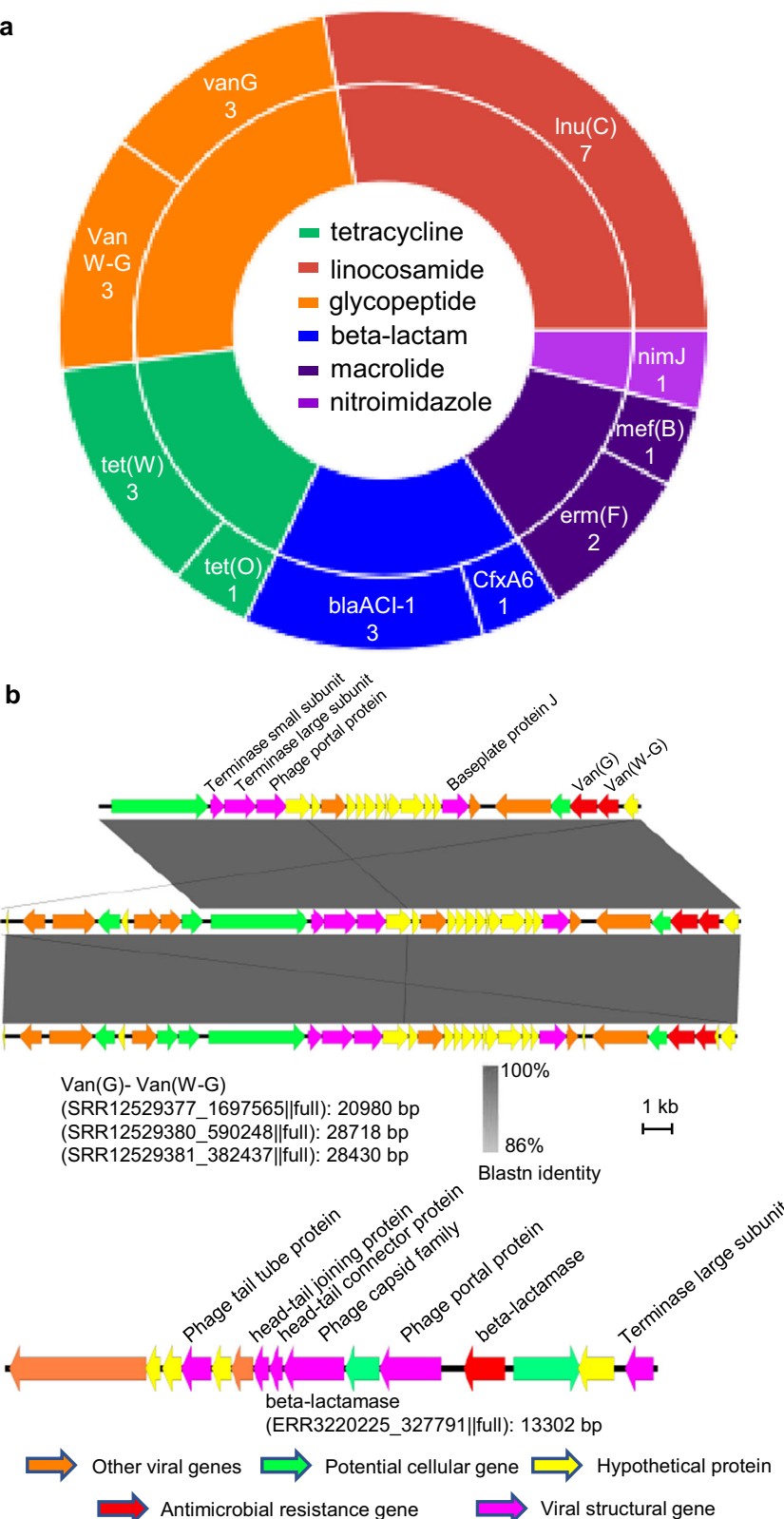

**Fig. 5 | ARG carried by rumen viruses. a** The number of ARG-carrying viruses and their ARG classes identified in the selected vial contigs (24 out of 705,380). **b** The genomic organization of three representative ARG-carrying viral contigs. These three viral contigs shared the same two *van* genes with highly similar sequences and had nearly identical genomic organization but were found in three different metagenomic samples. The gray lines connecting the genes among contigs indicate blastn identity. The contig at the bottom represents the viral contig with ARG flanked by viral hallmark genes. The representative viral contigs carrying other ARG genes were chosen based on CheckV completeness score and the number of cellular genes, and their genome organizations are displayed in Supplementary Fig. 6. The manual curations of each ARG-carrying contigs and their detailed annotation can be found in Supplementary Data 5.

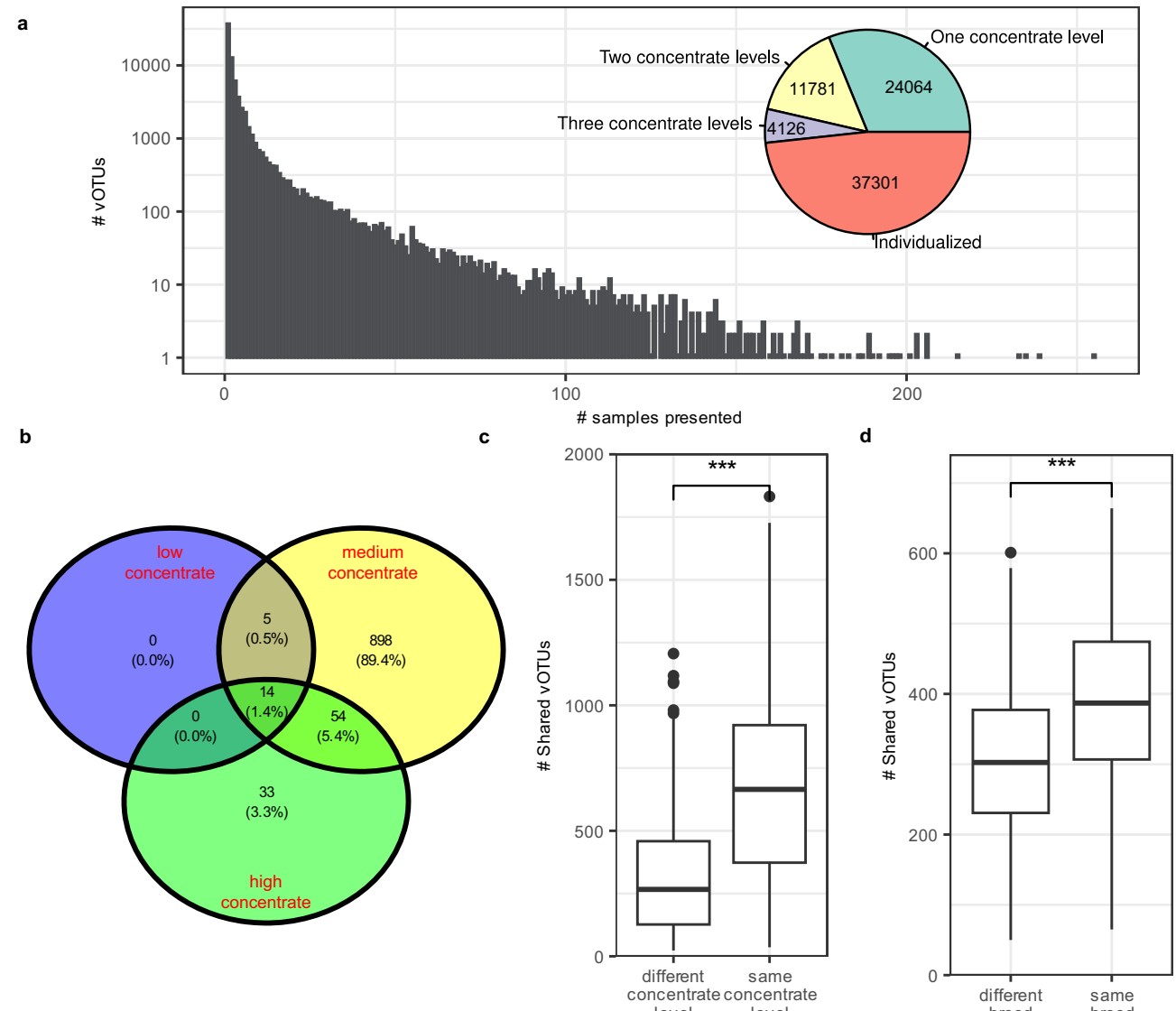

**Fig. 6 | Core virome of different animal species and husbandry regimens.**
**a** Numbers of vOTUs identified among 283 cattle fed one of three different levels of dietary concentrate: low (below 30%, n = 84), medium (30–50%, n = 80), and high (above 50%, n = 119) (bar plot), and the number of vOTUs observed in only one animals (individualized), more than one animals at one, two or three concentrate levels (pie chart). **b** A Venn diagram showing the number of vOTUs shared by more than 50% of the animals across fed diet with low (purple), medium (yellow) or high (green) concentrate. **c** The number of vOTUs shared between animals fed same (n = 445) or different (n = 416) levels of concentrate (data from ref. 80). **d** The

number of vOTUs shared between animals of different (n = 768) breeds or the same (n = 360) breed (data from ref. 79). The feeding management is the same for all samples. Box plots indicate the median (middle line), 25th and 75th percentile (box), and 5th and 95th percentile (whiskers) as well as individual observations (dots). Statistical significance was tested using the two-sided non-parametric Wilcoxon signed-rank test. P values below 0.001 was indicated as "***". Detailed information on the core vOTU (family-level taxonomy and predicted hosts) is shown in Supplementary Data 6.

metagenomes that we analyzed were derived from animals differing in genetics and physiology (different species and breeds), diet, husbandry (wild vs. domesticated, grazing vs. nongrazing), and geography. We did find a small core rumen virome among cattle fed diets with low, medium, or high levels of grain. Although the "size" of the core rumen virome varies depending on the homogeneity of the diet and probably also animal-specific factors such as genetics and age, the core rumen virome infects the core rumen microbiome, including common species that play important roles in rumen functions, such as species of *Butyrivibrio, Fibrobacter, Prevotella, Ruminococcus,* and *Methanobrevibacter*. Some of the species of these genera are indeed infected by bacteriophages isolated from the rumen[16,17]. Additionally, viruses can both destroy biofilms through predation and depolymerases[46] and can promote biofilm formation by spontaneously inducing prophages and releasing extracellular DNA as extracellular matrices for biofilm

formation[47]. By lysing fiber-degrading bacteria and affecting biofilms on feed particles, the rumen virome likely affects feed digestibility in a top-down manner, as previously proposed for the moose rumen virome, in which no GH-encoding AMGs were found[28]. Similarly, by lysing their host cells and increasing the amount of microbial protein available for degradation by proteolytic bacteria, rumen phages likely contribute to the intraruminal recycling of microbial protein[15], which decreases microbial protein outflow to the small intestine and nitrogen utilization efficiency in ruminants[48]. Therefore, the rumen virome probably also affects the supply of microbial protein to ruminants and thus nitrogen utilization efficiency in a top–down manner.

The phage infection of bacteria can lead to distinct virocell states that alter host metabolism, physiology, and ecology[7]. One underlying mechanism is the provision of hosts with AMGs that are involved in nutrient acquisition by and the metabolism of hosts. In other

ecosystems, AMGs have been found to impact several important ecological processes, including global carbon recycling[2], nitrogen metabolism in the ocean[49], and sulfur metabolism in the environment and the human gut[50]. The 41 categories of AMGs identified from 286 vMAGs, including 5 new categories, greatly expand the known repertoire of viral AMGs of the rumen ecosystem beyond the small numbers of AMGs previously reported in beef steers[29] and moose[28]. The fact that most of the categories were found previously in other ecosystems also indicates that the rate of false AMG annotation was low. Unlike the two studies that reported AMGs in the rumen viromes of steers[29] and moose[28], we found cellulase-encoding AMGs (GH5 and GH9) and showed that they encoded functional cellulases. Given the large number of incomplete viral contigs that were not subjected to AMG annotation and the fact that additional viruses remain to be identified, the actual rumen repertoire of viral AMGs is probably orders of magnitude larger than that revealed thus far. Nevertheless, the identification of AMGs involved in antiviral defense (e.g., DNMT1), polysaccharide digestion (e.g., GHs), ammonia-assimilation (e.g., asparagine synthase), and amino acid and vitamin biosynthesis (chorismate mutase and cobaltochelatase CobS) suggests that the rumen virome can also affect feed digestibility and microbial protein synthesis in a bottom-up manner. Future studies are warranted to determine how and to what extent the rumen virome affects feed digestibility and nitrogen utilization efficiency.

Unlike research on the human gut microbiome and virome, which mainly focuses on their involvement in health[51], research on the rumen microbiome and virome centers on their associations with feed efficiency and $CH_4$ emissions[52]. In most metagenomic studies, rumen bacteria, methanogens, protozoa, and fungi are analyzed, but rumen viruses/phages are often ignored due to the lack of reference databases and robust bioinformatics tools for identifying and classifying viral sequences from rumen metagenomes. As discussed above, rumen viruses can affect the rumen microbiome, major rumen functions, and animal performance, including feed efficiency and productivity (growth and lactation performance), in a multifaceted top-down and bottom-up manner. With RVD, future metagenomic research can analyze both the rumen microbiome and virome to better understand their roles in and associations with feed efficiency and animal productivity and to inform new approaches for improvement. Furthermore, the lytic phages predicted to infect undesirable rumen microbes, such as species of *Streptococcus*, rumen methanogens[53], and protozoa, and their enzymes may be isolated and explored for their potential to reduce rumen acidosis, methane emissions, and nitrogen excretion.

## Methods

### Assembly and identification of viral contigs from rumen metagenomes

Published rumen metagenomes (*n* = 975, Supplementary Data 1) were downloaded from NCBI SRA, subjected to quality control with fastp[54], and then assembled individually using Megahit (v1.2.1) with the default parameters. The obtained metagenomes were published in 80 studies covering 13 species or husbandry regimes of ruminants from 8 countries across 5 continents (Fig. 1). The countries where the metagenomes were sampled along with the numbers of metagenomes were visualized on a world map that we generated using the R package "rworldmap."

After assembly, tentative viral contigs were first identified following the viral sequence identification SOP (https://doi.org/10.17504/protocols.io.bwm5pc86). Briefly, tentative viral contigs >5 kb were verified using VirSorter2[22] (option: --min-score 0.5), and the contigs that passed the verification procedure were input to CheckV[21] to trim off host sequences flanking prophages. We only chose viral contigs >5 kb because the currently available bioinformatics tools show a relatively high false positive rate when identifying viral contigs <5 kb[30].

Only the contigs falling into categories Keep1 and Keep2 were retained as putative viral contigs (708,580 in total) for further analyses.

The viral contigs were then clustered into vOTUs according to 95% average nucleotide identity (ANI) across 85% of the shortest contigs as suggested[25] using a custom script from the CheckV repository[21]. The resultant 411,125 vOTUs were further verified with VIBRANT[23] (option: --virome). To be conservative, only the vOTUs identified by both VIBRANT and VirSorter2 (397,180) were used to build RVD and retained for taxonomic classification and host prediction. We chose to use both VIBRANT and VirSorter2 for viral identification because they are among the most recent tools with the best performance, as recently benchmarked[30]. No rumen viral database was available to aid in annotation, so we annotated the clean reads using complete genomes from NCBI RefSeq Viral (release 211; downloaded in March 2022) and the host-associated fraction of IMG/VR V3 database[27]. The completeness of the vOTUs was estimated using CheckV[21]. A rarefaction curve was generated to assess the coverage of the rumen viruses using the RTK[55] package in R.

### Taxonomic classification of all vOTUs and tree construction of those assigned to the order *Caudovirales*

We assigned vOTUs >10 kb to genus-level viral taxa based on a gene-sharing network using vConTACT2[26], which uses NCBI RefSeq Viral (release 88) as reference genomes. The vOTUs that could be clustered with the reference genomes of a viral genus were assigned to that genus according to the vConTACT2 workflow. We assigned the vOTUs that failed to be assigned to a viral genus and those <10 kb to family-level viral taxa using the majority rule, as applied previously[4]. Briefly, we predicted the ORFs of each vOTU using Prodigal[56] and then aligned the ORF sequences with those of NCBI RefSeq Viral using BLASTp with a bit score of ≥50. The vOTUs that were aligned with the NCBI RefSeq Viral genomes of a viral family with >50% of their protein sequences were assigned to that family. We identified crAss-like phages using BLASTn against 2,478 crAss-like phage genomes identified from previous studies[57–59], with a threshold of ≥80% sequence identity along ≥50% of the length of previously identified crAss-like vOTUs.

Most of the taxonomically assignable vOTUs of the rumen virome were assigned to the order *Caudovirales*, as in the case of the human gut virome, and we thus compared the phylogenic distribution of the *Caudovirales* viruses between the rumen and the human gut viromes based on concatenated protein phylogeny[60]. Specifically, we downloaded the HMM profiles of the 77 marker genes of *Caudovirales* viruses from VOGDB (http://vogdb.org) and searched RVD and the two largest human gut virome databases (MGV[5] and GPD[3], which phylogenetically complement each other) for the marker genes using HMMER v3.1b2[61]. To ensure a fair comparison across the databases, only the vOTUs with completeness >50% were included in the search. We then aligned each of the marker genes from the three databases using MAFFT[62], sliced out the positions with >50% gaps using trimAl[63], concatenated each aligned marker gene, and filled the gap where a marker gene was absent. Only the concatenated marker genes that each showed >3 marker genes and were found in >5% of all the aligned concatemers were retained, resulting in 10,203 *Caudovirales* marker gene concatemers, each with 13,573 alignment columns. These marker gene concatemers were clustered into genus-level vOTUs as described previously[5], where benchmarking was performed to achieve high taxonomic homogeneity using NCBI RefSeq Viral genomes. We built a phylogenetic tree of *Caudovirales* viruses using FastTree v.2.1.9 (option: -mlacc 2 -slownni -wag)[64] and aligned the concatenated marker genes of the representative vOTUs sequences of all the genus-level vOTUs with genome completeness >50% (based on CheckV analysis). The *Caudovirales* tree was visualized using iTOL[65]. The vOTUs identified as prophages or encoding an integrase were considered lysogenic. The lysogenic rate (%) was calculated based on the

VIBRANT results as the percentage of lysogenic viruses of all the viruses for each genus of their probable hosts.

## Host prediction and host phylogenetic tree construction

We predicted the probable hosts of the rumen viruses using an alignment-dependent method (aligning prophage sequences and CRISPR spacer sequences with genome databases of prokaryotes and rumen protozoa) with high prediction accuracy[66]. For prophage sequence alignment, we manually curated three genome databases: a database containing 22,095 bacterial MAGs and 410 archaeal MAGs assembled from the same metagenomes used for viral analysis in the current study, a database of 2,729 rumen prokaryotic MAGs of the Cow Rumen Genome Database V1.0 (https://www.ebi.ac.uk/metagenomics/genome-catalogues/cow-rumen-v1-0), and a database of 251,167 reference genomes of prokaryotes of NCBI RefSeq (release 211; downloaded in March 2022). The above prokaryotic MAGs and reference genomes were classified according to the GTDB taxonomy using GTDB-tk (option: -classify_wf)[67]. We did not predict the hosts of mycoviruses because most mycoviruses are RNA viruses, and only a few reference genomes of rumen fungi are available. We aligned the representative viral sequence of each vOTU with the above prokaryotic genome/MAG sequences using BLASTn (option: -task metablast) to identify integrated prophage regions. A host match was called when >2,500 bp of a host genome or MAG matched a vOTU sequence at >90% sequence identity over 75% of the vOTU sequence length[4]. We predicted probable protozoal hosts of the rumen viruses by searching the 52 high-quality ciliate SAGs[68] for EVEs using BLASTn and the above criteria.

We also predicted the probable hosts of the identified viruses by aligning the CRISPR spacer sequences of the vOTUs against the reference genomes and MAGs of prokaryotes. Briefly, we identified the CRISPR spacer sequences of the reference genomes and MAGs using MinCED (option: -minNR 2)[69] and then aligned the identified CRISPR spacer sequences with the sequences of RVD using BLASTn (option: -dust no). A probable host was called when the CRISPR spacer of a reference genome or MAG exactly matched a vOTUs sequence (100% identity and 100% coverage). In total, we identified 43,166 vOTUs that have a CRISPR spacer match, or the viral sequences are integrated into the host genomes. With the sequences of these vOTUs, we built a gene-sharing network using vConTACT2. After removing duplicated edges and clusters with <3 nodes, the network was imported into Cytoscape 3.7.2[70] and annotated based on the taxonomy of the viruses and their probable hosts.

To reveal the infection patterns of rumen viruses, we constructed genus-level phylogenetic trees for the identified hosts (archaea, bacteria, and ciliates). For the phylogenetic trees of bacterial and archaeal hosts, one genome was randomly chosen within each identified host genus. Then, the 120 marker genes of bacteria and 122 marker genes of archaea in the genomes of the selected bacteria and archaea were aligned using GTDB-tk[67]. Thereafter, phylogenetic trees were constructed using the aligned marker genes and IQ-TREE (option: -redo -bb 1000 -m MFP -mset WAG,LG,JTT,Dayhoff -mrate E,I,G,I + G -mfreq FU -wbtl)[71] and were visualized using iTOL[65]. Lysogenic rates were calculated based on the VIBRANT results as detailed above. A ciliate tree was acquired from Li et al. [68] and visualized using iTOL[65].

## Evaluation of RVD for utility in analyzing rumen viromes in previously published rumen metagenomes

We downloaded the publicly available metagenomic sequences of Anderson et al.[29] from NCBI SRA and identified the viruses using the same viral identification protocol mentioned above. That study used three library preparation and sequencing methodologies (sequencing of VLP-enriched rumen metagenomes with Ion Torrent PGM or Illumina MiSeq and sequencing of bulk rumen metagenomes with Ion Torrent PGM), and we compared the viral identification rate (percent of identified viral contigs over total contigs used for viral identification) among all three different methodologies. A two-sided Wilcoxon rank-sum test was used to compare the viral identification results in R. The viral contigs from VLP-enriched metagenomes and bulk metagenomes were clustered into vOTUs separately, and the resultant two sets of vOTUs were clustered with RVD using the clustering method described above. The percentage of lysogenic viral contigs was also calculated as detailed above. We compared the percentage of lysogenic viral contigs and the ratio of vOTUs clustered with RVD between the VLP-enriched metagenome and bulk metagenomes with the Chi-squared test in R. To test RVD for its utility for identifying viruses from metagenomes that were not used in the development of RVD, we also mapped the clean reads from previous rumen virome studies[28,29] to RVD using CoverM (option: --min-read-percent-identity 0.95, --min-read-aligned-percent 0.75, --min-covered-fraction 0.7; https://github.com/wwood/CoverM) with the trimmed mean method, which calculates the average coverage after removing the regions with the top 5% highest and lowest coverage.

We also evaluated RVD and the two largest virome databases, NCBI RefSeq Viral and IMG/VR V3 (the host-associated viruses only), for their utility in identifying rumen viruses from rumen metagenomes. Specifically, we mapped the sequencing reads of two recent sets of rumen metagenomes (BioProject PRJNA597489 and PRJNA779163) whose viruses are not included in any of the three virome databases. Then, we calculated the mapping rates of the viral sequences using CoverM and compared the mapping rates among the three virome databases. We also evaluated RVD in detecting viruses in rumen metagenomes from different ruminant species based on mapping rate. As high variability in the mapping rates was found, we only compared the results between healthy ruminants and those suffering from SARA. The mapping rates of viral reads were statistically analyzed using the two-sided nonparametric Kruskal−Wallis test in R.

## Identification of AMGs carried by the rumen vMAGs

We used stringent criteria to extract viral sequences, but during the initial manual curation of the viral contigs, we found some contigs that were possible host genomic islands. Such contigs can be misidentified as viral genomes by virus identification tools[3]. Additionally, it is still challenging to delineate the exact boundaries between host genomes and prophage genomes[21], and any remaining host genes, if not removed, can be misidentified as AMGs. Thus, we performed a series of curation and filtering steps to select vMAGs for AMG identification to minimize false AMG identification. First, we only searched the vMAGs >10 kb (5912 in total) for AMG identification using the criteria recommended in a benchmarking paper[30]. The selected vMAGs were then subjected to AMG identification and genome annotation using DRAMv[72] after processing with VirSorter2 with the options "−prep-for-dramv" applied. Second, the AMG-carrying vMAGs were removed if the AMGs were at an end of the vMAGs or if the AMGs were not flanked by both one viral hallmark gene and one viral-like gene or by two viral hallmark genes (category 1 and category 2 as determined by DRAMv). Third, the remaining vMAGs were further manually curated based on the criteria specified in the VirSorter2 SOP (https://doi.org/10.17504/protocols.io.bwm5pc86; also see https://github.com/yan1365/RVD/blob/main/vmags_check_helper/readme.txt). We eventually obtained 1,880 vMAGs. To further minimize false identification, we manually checked the genomic context of these vMAGs and found that some of them were still possible genomic islands. Therefore, we filtered the 1880 vMAGs based on the criteria established by Sun and Pratama et al. (unpublished data). Briefly, vMAGs with only integrases/transposases, tail fiber genes, or any nonviral genes were removed. The remaining vMAGs were filtered again to remove those that did not have at least one of the viral structural genes (i.e., capsid protein, portal protein, phage coat protein, baseplate, head protein, tail protein, virion structural protein, and terminase) and those containing genes encoding an

endonuclease, plasmid stability protein, lipopolysaccharide biosynthesis enzyme, glycosyltransferase (GT) families 11 and 25, nucleotidyltransferase, carbohydrate kinase, or nucleotide sugar epimerase. We eventually obtained 504 vMAGs free of genomic islands. To benchmark our curation pipeline, 100 of the vMAGs were randomly selected for detailed manual curation based on their genomic context. According to the benchmarking results, we were confident that we retained only complete vMAGs for AMG prediction. Detailed results of each curation step and full annotation of the final vMAGs and the annotation of the identified AMGs are presented in Supplementary Data 4. We compared the AMGs identified in the rumen virome to the previously identified AMGs from other viromes, which are available in an expert-curated AMG database (https://github.com/WrightonLabCSU/DRAM/blob/master/data/amg_database.tsv). For the newly identified AMGs, we double-checked the annotations and searched the literature to ensure that they were truly AMGs.

Cellulases are essential for feed digestion in the rumen, and we thus paid particular attention to vMAGs carrying cellulase-encoding AMGs. We found no cellulase-encoding AMGs from the vMAGs, and thus, we searched all the viral contigs in RVD. Following the manual curation steps described above, we identified two viral contigs, with one carrying a GH5 gene and the other carrying a GH9 gene. Although these two GH genes were not found in complete vMAGs, each was flanked by viral genes on both sides, indicating that they are part of the viral genomes. In addition, it should be noted that CheckV is database dependent, and both the genome database and the HMMs database used by CheckV contain very small numbers of rumen viruses. Therefore, it is reasonable to include "incomplete" viral contigs in the identification of AMGs, especially when the identified AMGs are carefully verified by manual curation.

### Cloning and expression of cellulase-encoding AMGs and enzyme activity verification

The DNA sequences of GH5 (1155 bp) and GH9 (2564 bp) were commercially synthesized by Synbio Technologies (Monmouth Junction, NJ, USA). The GH5 gene was double digested with BamH1 and Apa1, while the GH9 gene was digested with BamH1 and XbaI. Following gel purification, the GH5 and GH9 genes were cloned into *E. coli* DH5α (New England Biolabs, MA, USA) using pCDNA 3.1 (ThermoFisher Scientific, Waltham, MA USA). The successful cloning of the GH5 or GH9 gene was confirmed by isolating plasmid DNA from overnight *E. coli* cultures, subjecting plasmid DNA to double digestion with the restriction endonucleases mentioned above, and agarose gel (1.2%) electrophoresis.

The cellulase activity of the cloned GH5 and GH9 proteins was verified using both an agar plate assay[73] and the 3,5-dinitrosalicylic acid (DNS) reagent[74]. Briefly, overnight *E. coli* cultures carrying cloned GH5, GH9, or the empty cloning vector were inoculated on LB agar plates containing ampicillin (100 μg/ml) and CMC (0.5%). After incubation at 37 °C for 48 h, each plate was flooded with 5 ml of 0.2% Congo for 30 min. Each plate was then washed with distilled water and rinsed three times with 5 ml of 0.5 M sodium chloride for 15–20 min for color/halo development. The colonies showing large halos were further verified using the DNS agent. Briefly, *E. coli* cultures carrying the cloned GH5, GH9, and no insert (empty pCDNA 3.1) were grown in LB broth at 37 °C for 24 h and 48 h. Following centrifugation at $2000 \times g$ for 15 min at 4 °C, 500 μl of supernatant from each culture as the source of cloned GHs was mixed with 500 μl of 1% CMC (in 50 mM sodium acetate buffer, pH 5). The mixture was incubated at 60 °C for 30 min, and the reaction was stopped by adding 1.0 ml of DNS reagent and boiling for 5 min. Optical density was read at 540 nm, with an *E. coli* culture carrying empty pCDNA 3.1 as a blank. A standard curve was prepared with a series of glucose concentrations. The DNS assay was performed in triplicate (for both each *E. coli* culture and glucose standard).

### Identification of ARGs carried by the rumen viruses

The viral contigs were searched for ARGs using conservative criteria[37]. In brief, we downloaded the CARD database (v3.0.7)[75] and searched it for ARGs carried by the viral contigs of RVD using BLASTp with a threshold of 80% sequence identity and 40% alignment coverage. We also searched ARGs in RVD using the NCBI AMRfinder tools v.3.8.4[76], which is a highly accurate tool and uses an expert-curated built-in database. The identified probable ARG-carrying viral contigs were curated to retain bona fide viral sequences using the above pipeline to identify AMG-containing vMAGs. To be conservative, the viral contigs with ARGs at the end were removed unless they were adjacent to a virus-related gene. A total of 22 ARG-carrying viral contigs were found, and they were manually checked individually to determine their viral origin based on genome context annotation. The detailed annotation and manual curation of individual ARG-carrying viral contigs are shown in Supplementary Data 5. The representative viral contigs carrying each type of ARG were picked based on CheckV completeness and the number of cellular genes. The viral contigs with the highest completeness and least cellular genes were chosen as representative viral contig, and their genomic organizations were depicted individually using easyfig[77] and annotated manually.

### Distribution of different viral populations and ecological analysis

We evaluated RVD for the presence of a core rumen virome based on vOTU prevalence. We did not find a "global" core rumen virome across all 975 rumen metagenomes. We thus only focused on 283 cattle with reported dietary composition information. These cattle were divided into three groups based on the concentrate level of their diets: low, (below 30% concentrate), medium (30–50% concentrate), and high (>50% concentrate). First, we transformed the raw abundance table into a binary matrix (presence or absence). Then, the prevalence of each vOTU in each sample was calculated. A vOTU was included in the core rumen virome if its prevalence exceeded 50% of the prevalence for each concentrate level or all cattle. Based on prevalence, the vOTUs were categorized as individualized (observed in only one sample), one concentrate level (observed in more than 1 sample but exclusively from a single concentrate level), two concentrate levels (observed in animals from two concentrate levels) and three concentrate levels (observed in all three concentrate levels). The numbers of vOTUs shared by the core viromes among the three concentrate levels were visualized with a Venn graph in R. We examined whether animals from the same diet or same breed share more vOTUs compared to animals fed different diets or of different breeds using subsets of data from Stewart et al.[78] and Li et al.[79] respectively. The Kruskal–Wallis test was used to compare the numbers of shared vOTUs in different groups in R.

The coverage of each vOTU in the RVD in the 975 rumen metagenomes was further examined by mapping the metagenomic sequence reads to RVD using CoverM with the parameters described above. Based on the read mapping results, the viral richness per billion base pairs in each metagenome was calculated and used as the proxy for richness as described previously[4]. With the abundance table, beta-diversity was computed based on Bray–Curtis dissimilarity using the vegan package[80] in R, and PERMANOVA was performed with 999 permutations to test for differences among viromes with the adonis function of the vegan package in R. Viral richness was compared among different animal species, countries, and studies using the Kruskal–Wallis test in R. The results were visualized with ggplot2 in R. As study-by-study variation was seen in the alpha- and beta-diversity results, we tested how studies could be clustered with hierarchical clustering based on the number of vOTUs shared between studies as described previously[4]. For studies including multiple ruminant species or multiple production systems (dairy and beef), each species or system was considered a separate "study". Only studies each with >12

metagenomes were retained for the analysis. The number of vOTUs shared by two studies was compared for every study pair, and the results were subjected to hierarchical clustering. The hierarchical clustering results were visualized in R with the ComplexHeatmap package[81] and annotated according to the metadata.

## Statistics and reproducibility

The statistical tests were conducted in R and were as detailed in previous sections and in figure legends. The scripts for the statistical analysis and visualization are available in the GitHub repository (https://github.com/yan1365/RVD). No statistical method was used to predetermine sample size since no new experiments were conducted in this study. The data we used came from published studies and the treatment groups were determined based on the metadata reported by the individual studies.

## Reporting summary

Further information on research design is available in the Nature Portfolio Reporting Summary linked to this article.

## Data availability

The viral genomes included in RVD and the associated information can be accessed and downloaded without any restriction at https://zenodo.org/record/7412085#.ZDsE2XbMK5c.

## Code availability

The custom Bash, Python, and R scripts used to process and analyze the data and generate the figures are available at https://github.com/yan1365/RVD.

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

## Acknowledgements

The authors gratefully acknowledge funding for this project from the USDA National Institute of Food and Agriculture (Award number: 2021-67015-33393).

## Author contributions

M.Y. and Z.Y designed the study. M.Y. and A.A.P. performed the bioinformatic analyses. S.S. performed the cloning and expression of GH5 and GH9. Z.L. and Y.J. contributed to the bioinformatic analyses. M.Y. and Z.Y. wrote the manuscript. M.B.S. and all the other co-authors edited the drafts of the manuscript.

## Competing interests

The authors declare no competing interests.
