## [Peer Review File · Nature Communications]

REVIEWER COMMENTS

Reviewer #1 (Remarks to the Author):

This manuscript describes the compilation of a database including assembled genes and contigs of microbial viruses (predominantly phage populations) which have been pulled from publicly available metagenomes of the rumen. The study is unique in its approach to characterizing the viruses associated with rumen microbes, using many metagenomes and thus representing a significant computational effort. The several different bioinformatic techniques used were not novel or specifically designed for this study, but instead are previously published methods applied to the extensive dataset obtained.

While I found the manuscript to be interesting and several findings may be of interest to those interested in gut-associated microbial communities, including the rumen microbiologists, the manuscript was not of a quality that I would expect to be published in such a high impact journal as Nature Communications.

There have been other studies published describing the rumen virome (particularly those which physically enrich for the viral fraction of the rumen), and while the paper cites these articles, it essentially ignores the technical aspects and findings of previously published research and claims to be the first study of the rumen virome. This is incorrect and completely overstates the importance and significance of the study.

The English language used is very good but is sometimes written in a very casual style (not befitting a high impact journal), the sections are not distinct, with methodology described too much in both the results and discussion. Extensive revisions are required (too many to list). There is also a lot of discussion in the results section, which is very speculative in nature. While effort has been taken to extrapolate the findings into a microbial ecology and rumen-based context (e.g. significance of phage-mediated gene transfer within the rumen, including potential carbohydrate-degrading and antimicrobial resistance genes) these ideas are not well explained or compared well to previously published rumen virome and metagenome studies (e.g. the excellent article by Solden et al. 2018 <https://doi.org/10.1038/s41564-018-0225-4>). Some of the statements made e.g. viral communities associated with SARA conditions, would need extensive experimental work to validate. Are the increased viral populations identified actually an artifact of increased numbers of Proteobacteria and organisms such as *Streptococcus bovis*? Phages infecting *Streptococcus* are well characterized and present in current viral databases, so you are more likely to find them, rather than poorly characterized rumen microbial populations for which there are no previously published genomes (of infecting phages and prophage elements). IN addition, some bacteria which proliferate in high grain diets may be more readily infected by phages (*Streptococcus* readily pick up phages), this is one example, where there has been a lack of care and comparison with currently available literature.

There are also far too many figures included. I suggest that this manuscript be extensively revised and if experimental work to validate some of the claims is not possible then perhaps this manuscript should be submitted to a journal with a more bioinformatics focus.

Reviewer #2 (Remarks to the Author):

The manuscript "Unraveling the viral dark matter of the rumen microbiome with a new global virome database" submitted by Yan et al. described a rumen virome database using publicly available datasets. This is an interesting topic, however, more data are required before further evaluation.

1. Only 41,738 (out of 397,180) identified viral genomes have a completeness score higher than 50%. Is it a reliable virome database that we can use for future virome analysis?

2. Since the authors have built this database, the reviewer suggests that the authors should analyze the differences in the viral communities among different ruminants. Please pay more attention to the last part of the Results, and add more details.

3. It would be great if the authors could use this database to analyze several virus-like particle-enriched virome and total shotgun metagenome datasets, and compare the results to other databases or methods. This will contribute a lot to this manuscript.

4. How many complete viral genomes were used to predict AMGs? The frequency of AMGs and ARGs seems quite low. Does that mean the phages are less important than other mobile DNAs in the rumen microbiome regarding AMGs and ARGs?

Reviewer #3 (Remarks to the Author):

In this manuscript, the authors describe the development and evaluation of a novel rumen virome database mined from nearly 1000 rumen metagenomes collected and sequenced around the world.

Using current and conservative methods, the authors identified nearly 400,000 putative viral species, identified the carriage of antimicrobial resistance genes and auxiliary metabolic genes that could impact ruminant nutrition. They also demonstrated a 10-fold increase in the classification rate of viruses from rumen metagenomic data sets.

Although the work here is clearly communicated and well reported, the manuscript would benefit from the inclusion of the following:

1. Please include the specific VirSorter2 method described at the protocols.io link. In reviewing the content of the link, it appears that the methods described in the web resource are subject to change, and it is possible that they may not reflect the protocols used in this study if the link is accessed at dates into the future. I would encourage the authors to include this as a supplemental file so that the work is fully reproducible (by others) going forward.
2. Please augment figure 1C to reflect filtering/QC and assembly steps taken prior to the Virsorter2 analysis step. Although the virus-specific steps are highlighted in this figure, the sequence prep and assembly are not trivial and should be highlighted.

Please see below our point-by-point responses (shown in blue) to your concerns and comments.

REVIEWER COMMENTS

Reviewer #1 (Remarks to the Author):

This manuscript describes the compilation of a database including assembled genes and contigs of microbial viruses (predominantly phage populations) which have been pulled from publicly available metagenomes of the rumen. The study is unique in its approach to characterizing the viruses associated with rumen microbes, using many metagenomes and thus representing a significant computational effort. The several different bioinformatic techniques used were not novel or specifically designed for this study, but instead are previously published methods applied to the extensive dataset obtained.

AU: We appreciate your positive comments and the recognition of the unique aspects of this work. We also thank you for the constructive criticisms and suggestions that are very helpful for us to revise and improve the manuscript.

To minimize false identification of viruses, prediction of host, auxiliary metabolic genes, and antibiotics resistance genes, we used state-of-art bioinformatics tools and resources (the updated versions) that have been validated, benchmarked, and commonly used by other researchers. These bioinformatics tools result from more than a decade of innovative development and research on viromes in other microbiomes, community consensus from 6 international workshops, and lots of benchmarking efforts. This study represents an effort to apply these bioinformatics tools and analytical capabilities, with conserved and stringent criteria and cautions, to comprehensively curating and analyzing the rumen virome “hidden” in the rumen metagenomes.

While I found the manuscript to be interesting and several findings may be of interest to those interested in gut-associated microbial communities, including the rumen microbiologists, the manuscript was not of a quality that I would expect to be published in such as high impact journal as Nature Communications.

AU: We revised the manuscript substantially. We also added new results from additional analysis and lab experiments. Specifically, we explicitly described the evaluation RVD for its utility to identify viruses from metagenomic sequences, in comparison with IMG/VR, and to estimate virus abundance among different animal species and between health statuses (SARA vs. healthy animals) (Suppl. Fig. 2). Second, we calculated and compared the virus richness and beta diversity among animal species, geographies, and studies (Suppl. Fig 7). Third, we compared viral identification from metagenomes vs viral metagenomes (Suppl. Fig 3). Fourth, we experimentally verified whether two of the identified AMGs encode functional cellulases (one GH5 and one GH9) (Fig. 4c and 4d). Fifth, we examine study-by study difference in viromes for

possible drivers (Suppl. Fig. 8). Based on our results and results from previous studies, in the revised manuscript we added and emphasized new insights into rumen viromes in the Discussion, including 1) rumen virome may have profound impact on the rumen microbiome because the predicted hosts of core rumen virome are members of the core rumen microbiome (e.g., the genera *Prevotella*, *Butyrivibrio*, *Ruminococcus*, *Streptococcus*, *Methanobrevibacter*, *Methanosphaera*, and *Entodinium*); 2) rumen virome can play dichotomous roles in fiber digestion: enhancing fiber digestion by providing GHs-coding AMGs (especially cellulase genes) and biofilm formation on feed particles on the one hand, while on the other hand hindering fiber digestion by lysing fibrolytic bacteria (e.g., members of *Fibrobacter*, *Ruminococcus*, and *Butyrivibrio*) and destructing their biofilm on feed particles; 3) rumen virome likely affects nitrogen metabolism in the rumen, both positively and negatively with respect to nitrogen utilization efficiency: increasing microbial protein synthesis by providing AMGs encoding ammonia assimilation enzymes, while augmenting wasteful intraruminal cycling of microbial protein by lysis, decreasing microbial protein supply to animals; and 4) based on recent studies that demonstrated inhibition of *Methanobrevibacter* and *Streptococcus bovis* with viral enzymes, we discussed the potential to inhibit undesirable rumen microbes (methanogens, streptococci, and ciliates) to mitigate methane emissions, reduce rumen acidosis risk, and improve nitrogen utilization efficiency; and 5) rumen viruses likely impact the rumen microbiome and its functions in both a top-down (through lysis) and a bottom-up (through AMGs) manner.

There have been other studies published describing the rumen virome (particularly those which physically enrich for the viral fraction of the rumen), and while the paper cites these articles, it essentially ignores the technical aspects and findings of previously published research and claims to be the first study of the rumen virome. This is incorrect and completely overstates the importance and significance of the study.

AU: We apologize for not giving due diligence to some of the previous studies in the previous submission. We reread those papers and revised the manuscript by taking into account the results and conclusions of previous studies. Specifically, we highlighted the approaches, results, and contribution in the Introduction (lines 62-78) and discussed our results in the context of previous studies in the Discussion (lines 355-368, 373-381, 395-400).

We did mention “As the first comprehensive study on the rumen virome, we mined most of the published rumen metagenomes (nearly 1,000), recovered 397,180 species-level vOTUs, and created the first rumen virome database, RVD” in the Discussion. We rephrased that sentence and deleted the word “first”.

The English language used is very good but is sometimes written in a very casual style (not befitting a high impact journal), the sections are not distinct, with methodology described too much in both the results and discussion. Extensive revisions are required (too many to list).

AU: We extensively revised and edited the manuscript to improve clarity, flow, and coherence. We initially wanted to recapitulate some of the methodologies and precautions so that readers who do not read the Methods section could be informed about how the results were obtained and assess the quality of the results. That made the Results section quite long. In revising the manuscript, we deleted all the methodology recap except for the few that we think are needed to assess and understand the results. Just as examples, we delete the following:

“We predicted the hosts of the rumen prokaryotic viruses by matching the spacer sequences and prophage sequences, with conserved thresholds, with 251,167 reference genomes of prokaryotes in NCBI RefSeq (Release 211) plus 25,234 metagenome-assembled genomes (MAGs) of rumen prokaryotes (see Methods). Ciliate hosts were inferred using 52 single-cell amplified genomes (SAGs) of rumen ciliates⁴¹,”

“We generated a genome-based genus-level phylogenetic tree of the bacterial hosts (Fig. 3a) and the archaeal hosts (Fig. S2) with their predicted phages to examine the lysogeny rate, number of phages per genus of hosts, and number of phages per genome. Similarly, we also generated a genome-based phylogenetic tree of the 52 SAGs and their phages and estimated the number of phages carried by individual SAGs (Fig. S3)”

And moved the following to the Discussion and extensively revised them:

“Little is known about viruses infecting ciliates, but the large numbers of EVEs per rumen ciliate SAGs mirror the polyploidy and the huge number of chromosomes (>10,000) found in *Entodinium caudatum*⁴⁵, the most common rumen ciliate species. No EVEs have been documented in any ciliates, including model species *Tetrahymena thermophila*, *Paramecium aurelia*, and *Oxytricha trifallax*. But EVEs have been found in *Entamoeba* and *Giardia* in human stool metagenomes recently⁴⁶. Therefore, rumen ciliates probably carry EVEs.”

“Bacteroidota is the most abundant phylum in the rumen, and its species degrade and utilize major feed ingredients (hemicellulose, starch, and protein)⁴⁸. Rumen Bacteroidota is dominated by the genus *Prevotella*, which accounts for 40-60% of the total copies of bacterial 16S rRNA gene⁴⁹, while Methanobacteriota is dominated by the genus *Methanobrevibacter*, which can reach >63% of the total rumen methanogens⁵⁰.”

“By providing AMG encoding GHs and peptidases, rumen phages potentially enhance feed digestion. Synthesis of microbial protein from dietary nitrogen is the second most important function of the rumen microbiome, and interestingly one of the identified AMGs encodes

asparagine synthase, a key enzyme in ammonia assimilation by rumen bacteria⁵⁵. Taken together, the AMGs carried by some ruminal viruses can not only enhance their own survival but also affect diverse types of metabolism, such as metabolism of carbohydrates, nitrogen, and nucleotides; signaling; and transport (Fig. 4e).”

There is also a lot of discussion in the results section, which is very speculative in nature. While effort has been taken to extrapolate the findings into a microbial ecology and rumen-based context (e.g. significance of phage-mediated gene transfer within the rumen, including potential carbohydrate-degrading and antimicrobial resistance genes) these ideas are not well explained or compared well to previously published rumen virome and metagenome studies (e.g. the excellent article by Solden et al. 2018 <https://doi.org/10.1038/s41564-018-0225-4>).

AU: We moved the discussion from the Results to the Discussion, except for straightforward interpretation and extrapolation that is not discussed in the Discussion. We used the findings of previous studies on rumen viruses and virome, including the study by Solden et al. (reference 35), as the context in the Discussion.

We want to mention that the paper by Solden et al. reported a very good study on the rumen microbiome, but it only involved one animal, and rumen viruses were not the main focus. The authors did not find any AMGs that encode CAZymes and so they concluded that when moose are consuming a winter diet (primarily cellulose), rumen viruses are predominantly affecting carbon cycling in a top-down manner. We hesitated in comparing our results with theirs.

Some of the statements made e.g. viral communities associated with SARA conditions, would need extensive experimental work to validate. Are the increased viral populations identified actually an artifact of increased numbers of Proteobacteria and organisms such as *Streptococcus bovis*? Phages infecting *Streptococcus* are well characterized and present in current viral databases, so you are more likely to find them, rather than poorly characterized rumen microbial populations for which there are no previously published genomes (of infecting phages and prophage elements).

AU: We agree and acknowledged that more research would be needed to determine if rumen viruses are associated with SARA (lines 144-146). But the increase in viral abundance in both goats and dairy cows that suffer from SARA reasonably suggests that rumen phages might respond to SARA.

Proteobacteria, and in some cases, *S. bovis* too, increases in animals suffering SARA. It is plausible that their phages also increase, but that should not be considered artifacts. (As the host population of a phage increases, the phage increases also). It should be noted that we referred to the abundance of all viruses, both the viruses that infect well-characterized bacterial species and those that infect other bacterial species. Future research (such as studies comparing both the rumen viromes and microbiomes of the same groups of animals before and after SARA) is

needed to determine if SARA increases rumen viruses and if the increase is accompanied by an increase in their host, and this study will be useful.

In addition, some bacteria which proliferate in high grain diets may be more readily infected by phages (Streptococcus readily pick up phages), this is one example, where there has been a lack of care and comparison with currently available literature.

AU: Again we apologize for overlooking some previous work. In the revised manuscript, we included the findings of previous papers as background in the Introduction or as the context in the Discussion.

Some rumen bacteria, especially amylolytic bacteria, including species of *Streptococcus*, typically increase abundance when animals are fed high-concentrate diets. A number of phages infecting species of *Streptococcus*, including *S. bovis/equinus*, have been isolated and characterized. We've taken the results of a few related papers (references 28 and 29) as the context when we discuss our results in the Discussion (lines 357-363).

There are also far too many figures included. I suggest that this manuscript be extensively revised and if experimental work to validate some of the claims is not possible then perhaps this manuscript should be submitted to a journal with a more bioinformatics focus.

AU: Analyzing nearly 1,000 rumen metagenomes produced lots of data that are presented in many figures and tables. We did new analyses and cloned and expressed two cellulase-coding AMGs to verify their functionality. The data was included (please see our response to the second comment above). We moved Fig. 1c to a supplementary figure. In addition to 8 supplementary figures and 6 supplementary tables, this manuscript now has six figures. We think all the figures and supplementary figures and tables are needed to better understand the rumen virome from different aspects.

Reviewer #2 (Remarks to the Author):

The manuscript "Unraveling the viral dark matter of the rumen microbiome with a new global virome database" submitted by Yan et al. described a rumen virome database using publicly available datasets. This is an interesting topic, however, more data are required before further evaluation.

AU: We thank you and appreciate your positive comments and constructive criticism. We added more data, including experimental data on the functionality of two cellulase-coding auxiliary metabolic genes identified from the viruses (Figs 4c and 4d), comparison of viral enrichment on the detection of viruses (Suppl. Fig 3), richness and beta diversity of viruses across all the samples (Suppl. Fig. 7), and comparison of viromes among studies with respect to animal species, geographies, and sampling methods, and animal husbandry regimes. We also emphasize the evaluation of RVD for its utility to detect and identify viruses from metagenomic raw

sequences (lines 147-160). We extensively revised the manuscript and hope you will find the revisions address your concerns.

1. Only 41,738 (out of 397,180) identified viral genomes have a completeness score higher than 50%. Is it a reliable virome database that we can use for future virome analysis?

AU: This proportion is low but is relatively higher than that reported on human gut viromes in reference 6 (6.5%) and reference 4 (5.4%). The low proportions could be attributed to several factors. First, CheckV is a database-dependent bioinformatics tool, and it uses the genome databases currently available for quality filtering and completeness estimate. Thus, it works well to estimate completeness when the new sequence is closely related to a reference known sequence, but it does not work well for environments whose viromes are not yet known. Notably, the data in CheckV's databases are primarily derived from other ecosystems, not rumen. It is thus unclear how to interpret the completeness scores estimated by CheckV for the mostly novel viruses in the RVD. While CheckV completeness scores are low for many viromes, we feel that the databases are informative and useful for future virome analyses. As we noted in the manuscript, the ~400K viral genomes in RVD meet the community consensus requirement of Minimum Information about an Uncultivated Virus Genome (MIUViG). It shall be useful for future studies on rumen virome.

2. Since the authors have built this database, the reviewer suggests that the authors should analyze the differences in the viral communities among different ruminants. Please pay more attention to the last part of the Results, and add more details.

AU: We agree that comparison of rumen viromes between animal species and other factors that affect the rumen microbiome and animal performance (e.g., breeds, diets, animal husbandry regimes, animal performance, or health statuses) will provide new insights into the ecology of rumen viruses. But confounding factors will need to be eliminated, such as rumen sampling methods and DNA sequencing depth, both of which affect the detection of microbes and viruses. Doing so will increase the scope and amount of data. We are actually doing another virome study in which we take a more focused approach to examine how rumen virome is associated with animals (breeds), nutrition (diet, rumen fermentation profiles, feed efficiency), and animal productivity (weight gain, lactation performance).

The analysis for core virome involves a comparison of the rumen viruses across all the animals, species, or animal husbandry regimes. We added more details on the core virome (lines 279-290). We also compared the alpha diversity (species richness) and beta diversity in the revised manuscript (Suppl. Fig. 7) which showed large variations among studies/projects. The "study effect" was further examined and the results were included (Suppl. Fig. 8).

3. It would be great if the authors could use this database to analyze several virus-like particle-enriched virome and total shotgun metagenome datasets, and compare the results to other databases or methods. This will contribute a lot to this manuscript.

AU: Using RVD we analyzed the viruses captured in both viral metagenomes (virions enriched) and metagenomes (virions not enriched) reported in the rumen of beef steers (reference 34). We chose the data from that study because the viral metagenomes and metagenomes were prepared from the same rumen samples, eliminating confounding factors (e.g., diet, animals). We found that viral metagenomics with enriched virions increases the detection of viral sequences but does not affect the proportion of lytic viruses detected (Suppl. Fig. 3).

We also tested RVD in detecting and identifying viruses from the raw metagenomic sequences derived from beef steers and moose (reference 35). (The viral contigs assembled by the authors were not found in public databases). RVD allowed the detection of viruses at the species level, assignment of some vOTUs to known viral genera, and host prediction at the species level, rather than at the phylum level (reference 34) or MAG level (reference 35). The results are presented in the second section of the Results (lines 147-160).

4. How many complete viral genomes were used to predict AMGs? The frequency of AMGs and ARGs seems quite low. Does that mean the phages are less important than other mobile DNAs in the rumen microbiome regarding AMGs and ARGs?

AU: We predict AMGs from 5,912 complete viral genomes, 504 of them were predicted to carry AMGs (~10%). To minimize false prediction, we only predicted AMGs from highly certain viral genomes with stringent criteria. Still, we identified diverse AMGs (62 different categories). More categories and a higher prevalence of viral AMGs were found in the current study compared to the other two studies that reported AMGs in the rumen (references 34 and 35). It is more likely that viral AMG prevalence was not underestimated even though more AMGs will be identified (because the large number of incomplete viral genomes were not subject to AMG detection and more viruses remain to be identified). We discussed the potential that viral AMGs could affect the rumen microbiome in a bottom-up manner (lines 376-384).

ARGs are sparse in the rumen, and this concurs with the sparsity of viral ARGs in the human virome (reference 4) and in sequenced phage genomes (reference 49). It is an interesting question whether phages carry and spread more, or less, ARGs than the other mobile DNA elements (e.g., plasmids, transposons), but we found no information in the literature. We discussed the potential of rumen viruses in ARG dissemination (lines 270-272).

Reviewer #3 (Remarks to the Author):

In this manuscript, the authors describe the development and evaluation of a novel rumen virome database mined from nearly 1000 rumen metagenomes collected and sequenced around the

world. Using current and conservative methods, the authors identified nearly 400,000 putative viral species, identified the carriage of antimicrobial resistance genes and auxiliary metabolic genes that could impact ruminant nutrition. They also demonstrated a 10-fold increase in the classification rate of viruses from rumen metagenomic data sets.

Although the work here is clearly communicated and well reported, the manuscript would benefit from the inclusion of the following:

1. Please include the specific VirSorter2 method described at the protocols.io link. In reviewing the content of the link, it appears that the methods described in the web resource are subject to change, and it is possible that they may not reflect the protocols used in this study if the link is accessed at dates into the future. I would encourage the authors to include this as a supplemental file so that the work is fully reproducible (by others) going forward.

AU: We uploaded the protocols we followed as a readme file along with the source codes at GitHub (https://github.com/yan1365/RVD/tree/main/vmags_check_helper/).

2. Please augment figure 1C to reflect filtering/QC and assembly steps taken prior to the Virsorter2 analysis step. Although the virus-specific steps are highlighted in this figure, the sequence prep and assembly are not trivial and should be highlighted.

AU: The QC and assembly steps were added as suggested (now Suppl. Fig. 1).

REVIEWER COMMENTS

Reviewer #1 (Remarks to the Author)

This manuscript has been extensively revised and has taken into consideration the comments made by the initial three reviewers. In this regard the manuscript is now much improved, with many edits being made. The writing style is less casual, the methods section in particular is much more comprehensive and the significance of findings are not so overstated. The authors have also included additional bioinformatic analysis and some laboratory validation i.e., the expression and testing of two carbohydrate-degrading enzymes identified within the extensive database of viral genes.

The editing of the manuscript, while commendable, has introduced some additional errors. The language and grammar used has been extensively revised, however there are still some grammatical errors, missing words (“the” is often omitted, even in the abstract) and some overly dramatic words (drastic, ‘bona fide’). The introduction is still lacking in detail, referring to review articles rather than original citations. Although there are gaps in our knowledge of rumen viral populations, they have been extensively studied in the past, with studies were often limited by the techniques available at the time. These early studies identified that the rumen contains dense and diverse populations of viruses, predominantly tailed phages classified in the Caudovirales (Siphoviridae, Myoviridae and Podoviridae), as well as some other morphotypes (Inoviridae etc). The current study has also shown exactly this, it has just used sequence-based technologies to show that there is a lot of diversity in the viral populations, which can now be classified to lower taxonomic levels, although the vast majority are still classified in the tailed phage, Caudovirales families (Siphoviridae, Myoviridae and Podoviridae). The ‘big picture’ is essentially the same. In this regard, the introduction still downplays the scientific achievements of previous studies and relies on including citations which are actually reviews, rather than original articles. Several errors have been introduced (see list below). I am very familiar with some of the articles cited, and that the seven authors did not pick up the error in line 66, where CRISPR/cas sequences are noted to be present in rumen phage genomes, is an unfortunate oversight. CRISPR/cas systems are found in the genomes of bacteria and archaea (potential phage hosts), where they act as phage defence systems. They are not present in phage genomes.

The results section has been extensively revised and now contains less superfluous details about the methodology used. The wording regarding “enriched” (presumably viral metagenomes derived from viral particle fractions of rumen fluid) and metagenomic studies (metagenomes derived from DNA extracted from whole rumen fluid, hopefully microbial and viral material, not also including plant material), is not well defined. This comment applies throughout the manuscript and at times makes explanations confusing.

The results section still contains a lot of discussion, which should not be included as results. The point about single stranded DNA viruses, which I understood could be over-amplified in metagenomics studies, needs clearer explanation (the Nayfach et al. 2021 article explains this very

well). I also didn't understand some points mentioned in the results, for example the proportions of lytic vs lysogenic viral sequences identified (Supplementary Fig 3 is just confusing, even including some percentage differences would help). The phage-host interactions described are also confusing and perhaps influenced by using so many metagenomes from different animals and on different diets? I also think the numbers of different viruses included in the database needs to given perspective. The rumen fluid collected from most cattle will not contain highly abundant populations of 25 different genera of archaea. Depending on the diet, they mainly contain *Methanobrevibacter*, in relatively low proportions compared to the overall rumen bacterial populations. Individual cattle also won't contain over 2,400 different types of archaeophage. This high number is an artifact of pooling so many different metagenomes.

The database this study has generated is a useful tool for rumen microbiologists and possibly for other gut microbiologists. I am concerned however, that some of the "findings" reported in the current manuscript still overstate and extrapolate too far, without additional validation and without consideration for the metadata behind the metagenomes examined.

The concept of a "core" rumen virome is good, however using metagenomes from animals of different ruminant species and on vastly different diets isn't ideal, it would be more useful to have the core for specific ruminant species and dietary systems reported. As the authors note, the composition of the rumen microbiome is primarily determined by the diet of the animal, with particularly bacterial and archaeal populations fluctuating in response to the dietary substrates available. As rumen viral populations are entirely dependant on their microbial hosts for their proliferation, the viral populations also vary in response to diet (reported in several pre-genomics studies, as well as the Anderson paper), therefore generating a "core" virome from datasets derived from ruminants on all kinds of diets, is of questionable accuracy or value. I realise the authors investigated the "study" effect, but if this effect was significant or notable (which I understand it was), then surely there should not be so much emphasis on the "findings".

There is also a lot of discussion about auxiliary genes carried by viruses, but some of the genes noted, are surely just normal phage and viral genes? All viruses apart from the very large viruses, are compact in nature, encoding structural and non-structural proteins. These aren't auxiliary, they are required for normal viral function, particularly the nucleic acid replication proteins (which were noted in this study). This results/discussion point need extensive revision and clarification. Some of the genes listed in Figure 4 really aren't auxiliary genes. There has been an accumulation of different annotations in e.g. the ncbi nr protein database, for genes with similar function. The phage lysin genes (e.g. endoisopeptidases, N-acetylmuramoyl-L-alanine amidases) are a good example of this and even though they are classified within the CAZy GH25, they are often not regarded as glycosyl hydrolases in the literature. Manual annotation and nomenclature checks are required.

Similarly, the comments about antimicrobial genes were unwarranted without a thorough consideration of the metadata behind the metagenomics studies used to generate this virome dataset/database. Due care should always be given in the scientific literature to statements about the use of antimicrobials in animal production, as some countries have much greater regulation regarding antibiotic use than others. This study took data from animal studies conducted in many countries, and unqualified, over-generalised statements about antibiotic use in agriculture and

impact on the food chain is not appropriate and should be avoided. Omit (or extensively revise) the paragraph from lines 255-272.

The manuscript submitted for this second revision still requires extensive revisions.

Reviewer #2 (Remarks to the Author):

The revised manuscript "Unraveling the viral dark matter of the rumen microbiome with a new global virome database" by Yan et al. has incorporated additional analyses and experiments. However, the manuscript still lacks substantial innovation, has a limited research scope, and predominantly provides descriptive results that occasionally lack precision.

Here are some major points:

1. One weakness of this study is the lack of sufficient data to substantiate the quality of the proposed database. The analysis in Line113 is deemed inaccurate as the authors only utilize a subset of the datasets to validate the specificity of the database, which is derived from all the datasets. Consequently, the reported 10-fold expansion of rumen viruses may be overestimated. Additionally, the analysis in Line139 primarily focuses on host prediction, yet phage host prediction is a complex task that cannot be adequately resolved using a single phage database.
2. Line147-160: One aspect that needs improvement in this study is the clarity of the provided information. For instance, the statement "The authors could only assign the 2,243 viral contigs to families using BLASTn against NCBI RefSeq Viral" suggests that the cited study was able to assign a greater number of viruses compared to RVD. The writing is difficult to understand here. This part cannot be evaluated.
3. Line 161: This does not seem the right way to compare VLP-enriched and shotgun metagenomic data. The big limitation of vOTU derived from metagenome is that, prophages, even defective ones that are not viruses, cannot be distinguished. This is the main cause of the results here. The data actually suggested real viral genomes have been overestimated in RVD.
4. Line171-184: The writing is difficult to understand here. The term "Genus" and "Species" are very confusing. This part cannot be evaluated.
5. Line192: "Lysogeny rate" is not a proper term, but extensively used in the manuscript. The method used to calculate "Lysogeny rate" is not clear.
6. Line248: The new experiments did not contribute much to the manuscript. The authors are validating some known gene functions.

7. Line274: The reviewer expected a comprehensive analysis of virome in different ruminants. The “core” virome analysis did not contribute too much to the manuscript.

8. The manuscript primarily presents descriptive results without a well-defined scientific narrative, and the innovative aspects of the bioinformatic methodology employed are limited.

9. The writing is not in a proper way. Too many citations in the results, which made it difficult to read. The total number of references is almost 100, which is too much for a research article.

Reviewer #3 (Remarks to the Author):

In this manuscript, the authors have provided substantially reorganized and revised version of their prior submission. While still focusing on the use of a globally-distributed collection of rumen metagenomes for the discovery of viruses, the authors have provided additional detail, context, and experimentally demonstrated activity of a viral auxiliary metabolic gene (cellulase). These updates represent a substantial improvement to the manuscript and would allow the work to be fully replicated by others.

REVIEWER COMMENTS

Reviewer #1 (Remarks to the Author) --See also attachment---:

This manuscript has been extensively revised and has taken into consideration the comments made by the initial three reviewers. In this regard the manuscript is now much improved, with many edits being made. The writing style is less casual, the methods section in particular is much more comprehensive and the significance of findings are not so overstated. The authors have also included additional bioinformatic analysis and some laboratory validation i.e., the expression and testing of two carbohydrate-degrading enzymes identified within the extensive database of viral genes.

AU: Appreciate your recognition of the revision and improvement.

The editing of the manuscript, while commendable, has introduced some additional errors. The language and grammar used has been extensively revised, however there are still some grammatical errors, missing words (“the” is often omitted, even in the abstract) and some overly dramatic words (drastic, ‘bona fide’).

AU: We edited the manuscript extensively. We also have the manuscript edited by SpringerNature to correct grammatical errors.

The introduction is still lacking in detail, referring to review articles rather than original citations. Although there are gaps in our knowledge of rumen viral populations, they have been extensively studied in the past, with studies were often limited by the techniques available at the time. These early studies identified that the rumen contains dense and diverse populations of viruses, predominantly tailed phages classified in the Caudovirales (Siphoviridae, Myoviridae and Podoviridae), as well as some other morphotypes (Inoviridae etc.). The current study has also shown exactly this, it has just used sequence-based technologies to show that there is a lot of diversity in the viral populations, which can now be classified to lower taxonomic levels, although the vast majority are still classified in the tailed phage, Caudovirales families (Siphoviridae, Myoviridae and Podoviridae). The ‘big picture’ is essentially the same. In this regard, the introduction still downplays the scientific achievements of previous studies and relies on including citations which are actually reviews, rather than original articles.

AU: We are aware of most of the previous studies on the rumen phages (>20). We cited reviews and some previous studies pertinent to the main objective and scope of the current study to

provide relevant background and justification. Each study added new knowledge to the accumulating body of literature. We cited representative studies that made important contributions to the understanding of the rumen viruses, such as the early discovery of rumen phages, culture-based studies that demonstrated viral infections, association with animals or diets, and discovery of AMGs (lines 50-74, 79-84, 279-282, 306-308, 320-322). We do not intend to downplay the contribution of any previous studies.

Several errors have been introduced (see list below). I am very familiar with some of the articles cited, and that the seven authors did not pick up the error in line 66, where CRISPR/cas sequences are noted to be present in rumen phage genomes, is an unfortunate oversight. CRISPR/cas systems are found in the genomes of bacteria and archaea (potential phage hosts), where they act as phage defence systems. They are not present in phage genomes.

AU: Apology for the oversight. The error has been corrected.

The results section has been extensively revised and now contains less superfluous details about the methodology used. The wording regarding “enriched” (presumably viral metagenomes derived from viral particle fractions of rumen fluid) and metagenomic studies (metagenomes derived from DNA extracted from whole rumen fluid, hopefully microbial and viral material, not also including plant material), is not well defined. This comment applies throughout the manuscript and at times makes explanations confusing.

AU: We rephrased the wording according to the definition of previous publications (References 4 and 5). Specifically, we changed “enriched metagenome” to “viral-like particles (VLP) enriched metagenomes” and changed “metagenomes” to “bulk metagenomes”. Changes were made throughout the manuscript.

The results section still contains a lot of discussion, which should not be included as results.

AU: We moved the discussions in the Results to the Discussion section, except for the very brief and straightforward interpretations.

The point about single stranded DNA viruses, which I understood could be over-amplified in metagenomics studies, needs clearer explanation (the Nayfach et al. 2021 article explains this very well).

AU: More information was added to clarify that.

I also didn't understand some points mentioned in the results, for example the proportions of lytic vs lysogenic viral sequences identified (Supplementary Fig 3 is just confusing, even including some percentage differences would help).

AU: The percentages were added.

The phage-host interactions described are also confusing and perhaps influenced by using so many metagenomes from different animals and on different diets?

AU: Any factors, including animal or diet, that affect the rumen microbiome could affect the virome. Our main objective is not to examine the phage-host interaction in a specific context of animal or diet but to predict the probable host that the identified phages might infect. We think the predicted hosts of the phages can help future hypothesis-driven studies to investigate how phages interact with their host in a specific group of animals fed a given diet.

We used the bioinformatic methods commonly used in previous virome studies to infer probable hosts. In the Discussion, we pointed out that host prediction was performed based solely on bioinformatic analysis and caution should be taken when interpreting the results.

I also think the numbers of different viruses included in the database needs to given perspective. The rumen fluid collected from most cattle will not contain highly abundant populations of 25 different genera of archaea. Depending on the diet, they mainly contain Methanobrevibacter, in relatively low proportions compared to the overall rumen bacterial populations.

AU: We agree that many of the genera as probable phage hosts may not be abundant in a given group of animals, but we analyzed nearly 1,000 metagenomes from different animal species fed different diets across many countries. The large number of phages identified reflects the large number of metagenomes. The large number of genera of bacteria and archaea as probably hosts

is consistent with the large number of phages found. The prediction of a host does not imply it is abundant. It only shows it is a probable host of a phage.

We also want to point out that many previous genera or unclassified groups in NCBI or Silva have each been placed into multiple genera in the GTDB, which was used in this study. For example, the genus *Methanobrevibacter* has been split into five new genera in GTDB: *Methanobrevibacter*, *Methanobrevibacter_A*, *Methanobrevibacter_B*, *Methanobrevibacter_C*, and *Methanobrevibacter_D*. Many bacterial genera have also been split into multiple genera. For example, the genus *Ruminococcus* is now split into seven genera in GTDB. The large number of genera of bacteria and archaea as probable hosts also reflects the expansion/refinement of the genome-resolved taxonomy in GTDB.

Individual cattle also won't contain over 2,400 different types of archaeophage. This high number is an artifact of pooling so many different metagenomes.

AU: That number refers to all the different archaeophages identified in the rumen ecosystem represented by the nearly 1,000 rumen metagenomes we analyzed, not the number of different archaeophages from any individual animals. Actually, the archaeophages detected in individual cattle are very low (results not shown in this study). The number of different archaeophages we identified is reasonable considering 1,279 archaeal viral species were found in the human gut (<https://www.nature.com/articles/s41467-022-35735-y>) that has a much lower diversity and abundance of methanogens.

The database this study has generated is a useful tool for rumen microbiologists and possibly for other gut microbiologists. I am concerned however, that some of the “findings” reported in the current manuscript still overstate and extrapolate too far, without additional validation and without consideration for the metadata behind the metagenomes examined.

AU: We appreciated your constructive comments. Our main objective was to 1) create a database of the rumen virome by analyzing the occurrence and diversity of viruses in a large number of rumen metagenomes from a wide range of animals as a resource for future studies and 2) predict the probable host that the viruses/phages may infect to infer the potential role of rumen viruses on rumen microbiome and functions. We think that RVD and the findings of this study can help

future studies to better investigate the association or interaction of rumen viruses with animals, diet, and rumen microbiome. We revised the manuscript to avoid such misleading.

The concept of a “core” rumen virome is good, however using metagenomes from animals of different ruminant species and on vastly different diets isn’t ideal, it would be more useful to have the core for specific ruminant species and dietary systems reported. As the authors note, the composition of the rumen microbiome is primarily determined by the diet of the animal, with particularly bacterial and archaeal populations fluctuating in response to the dietary substrates available. As rumen viral populations are entirely dependant on their microbial hosts for their proliferation, the viral populations also vary in response to diet (reported in several pre-genomics studies, as well as the Anderson paper), therefore generating a “core” virome from datasets derived from ruminants on all kinds of diets, is of questionable accuracy or value. I realise the authors investigated the “study” effect, but if this effect was significant or notable (which I understand it was), then surely there should not be so much emphasis on the “findings”.

AU: We appreciate the comment. We agree the size of the core will vary depending on the types of animals and diets included. We repeated the analysis for the core virome but focused only on 283 cattle samples with reported dietary information. We grouped the 283 cattle samples based on the levels of dietary concentrate (low, medium, and high) and examined the vOTUs shared by samples within each concentrate level to determine the “core” virome at each concentrate level. Consistent with our previous analysis, the rumen virome profile is highly individualized. Considering the small sample size at each concentrate level, we did not emphasize the “core” for each concentrate level, but 14 vOTUs were found to be “core” across all three concentrate levels. 13 of them were predicted to infect *Prevotella* species. This indicates that at least some phages of the core rumen virome can infect the species of the core rumen microbiome. The new analysis results are included in the revised manuscript.

Since diet and animal genetics are two major factors affecting the rumen microbial profiles, we decided to test whether it is also true for the rumen virome profiles. We hypothesized that, although rumen virome profiles are highly individualized, more vOTUs would be shared by animals of the same breed or fed the same diet compared to animals of different breeds or fed

different diets. We tested the hypothesis using data from two studies (References 80 and 81). The results support the above hypothesis. The new results were added to the revised manuscript.

There is also a lot of discussion about auxiliary genes carried by viruses, but some of the genes noted, are surely just normal phage and viral genes? All viruses apart from the very large viruses, are compact in nature, encoding structural and non-structural proteins. These aren't auxiliary, they are required for normal viral function, particularly the nucleic acid replication proteins (which were noted in this study). This results/discussion point need extensive revision and clarification. Some of the genes listed in Figure 4 really aren't auxiliary genes. There has been an accumulation of different annotations in e.g. the ncbi nr protein database, for genes with similar function. The phage lysin genes (e.g. endoisopeptidases, N-acetylmuramoyl-L-alanine amidases) are a good example of this and even though they are classified within the CAZy GH25, they are often not regarded as glycosyl hydrolases in the literature. Manual annotation and nomenclature checks are required.

AU: We agree viral genomes carry many genes that are involved in functions beyond genome replication and synthesis structural components, such as genes encoding endolysin and holin, and should not be considered AMGs. We used DRAMv, which is an annotation tool predicting potential AMGs using a set of defined rules, and caution manual curation to ensure the vMAGs carrying potential "AMG" are truly vMAGs. To further minimize false AMG prediction, we did another round of manual curation focusing on the newly identified AMGs, checked the annotation and nomenclature of each AMG using the annotations from multiple databases (KEGG, Pfam, dbCAN, MEROPS, and virus-specific database VOGDB and RefSeq Viral), and searched the literature for the newly identified AMGs. To be more conservative, we excluded those that we are not completely sure about and only kept 5 out of the 22 newly identified AMG (found only in the rumen, lines 222-238).

Similarly, the comments about antimicrobial genes were unwarranted without a thorough consideration of the metadata behind the metagenomics studies used to generate this virome dataset/database. Due care should always be given in the scientific literature to statements about the use of antimicrobials in animal production, as some countries have much greater regulation regarding antibiotic use than others. This study took data from animal studies conducted in many

countries, and unqualified, over-generalized statements about antibiotic use in agriculture and impact on the food chain is not appropriate and should be avoided. Omit (or extensively revise) the paragraph from lines 255-272.

AU: We agree the use of antibiotics varies greatly between regions and animal husbandries. We repeated the analysis by focusing on a narrow scope (lines 249-257): First, we focused only on feedlot and dairy farm cattle and included the ‘region effect’ by analyzing data from China, North America (USA and Canada), and the UK. In the UK and North America, the results were consistent with our previous results. However, for China, no difference was seen between dairy and beef cattle, which probably reflects the less regulated antibiotic use in China. Second, we compared ARG-carrying virus abundance between beef and dairy cattle in different regions. For dairy cattle, China has the highest ARG-carrying virus abundance, which in general supports the previous results.

The manuscript submitted for this second revision still requires extensive revisions.

AU: In addition to revisions to address the comments and concerns from the reviewers, with help from SpringerNature Author Services, we also revised and edited the manuscript to improve clarity and readability.

Additional comments:

How do you spell unravelling? The title might need correction. (I thought it was two L’s).

AU: Both are correct. “unravelling” is used in British English, while “unraveling” is used in American English. But we replaced “unraveling” with “Elucidating” for clarity.

There some overly dramatic and repeated words (e.g. drastic, drastically), ‘bona fide’ (is there is non- bona fide list somewhere?) dotted throughout the text, please remove these. There was also a comment about viruses being ‘hidden’ which I don’t agree with, it is well known that microbial ecosystems (including the rumen) always contain viral (usually phage) populations.

AU: The overly dramatic words are replaced.

The abstract needs rewording in places (good proof-read), there are missing words (e.g. the).

AU: The grammar errors were corrected.

I have focused mainly on the Introduction section, although some comments apply throughout the text.

Line 41, missing words? Databases they generated,

AU: Revised.

Line 43 ‘leaving it poorly characterised and understood’ is a bit harsh, reword, this study addresses a knowledge gap.

AU: Rephrased.

Line 46 “and growing themselves’, omit

AU: Rephrased

Line 51, the citation here for the viral count is actually a very general review article – include original references

AU: Replaced with the original research article.

Line 51, “poorly understood” repeat of line 43.

AU: Revised.

Line 53 (now line 51), “estimated bacteriophage species to be <40 per rumen’, this is incorrect, it has been oversimplified from a review article. I checked this article, it’s not ‘species’, it’s 26-40 morphological types (in two TEM studies), and considering the majority of the phages in the rumen are Caudovirales (tailed phage morphotypes), this number of morphotypes is not unreasonable.

The Letarov article noted the number of detected phage morphological types, which “varies significantly between studies. Paynter et al. (1969) reported only six different phage types in bovine rumenal fluids, compared to 26–40 types found by other authors (Ritchie et al. 1970; Klieve and Bauchop 1988).”

AU: Sorry for the overlook, it was revised.

I agree, reliance on morphology-based classification is no longer recognised, but it was an important tool used for many years.

AU: The contribution of morphology-based classification work is recognized (lines 50-56).

Line 63 (now line 62), I think there are 10 rumen phages genome sequenced – check this.

AU: Revised.

Line 65, check wording, it is the bacterial hosts which play a role in feed digestion, not the phages. Line 68, remove “those”

AU: Revised.

Line 68, remove “drastically”

AU: Revised.

Lines 72 to 78, there is a table (Table 2 DOI: 10.3389/fmicb.2020.00450) listing the majority of rumen virome studies – use this if you like to update this paragraph. You only mention two studies, some of the sheep and dairy cow studies out there used many more animals, I feel the results mentioned are an underestimate. As sequencing technology has changed, the depth to which you can characterize the rumen virome has increased. Perhaps the authors are unaware, the number of animals you can use in a study is regulated by budgets (working with animals is expensive) and animal ethics concerns (use of animals for the purposes of experimentation is, and should be, highly regulated). Often animal numbers included in a study can't be increased, so the importance of these previous studies should not be downplayed. Expressions like “only a teaser” should be omitted.

AU: We saw that table. However, instead of citing all of them, we decided to only discuss those directly related to the rumen (Dinsdale et al., 2008 and Willner et al., 2009 only included some rumen samples, and they did not have any result/conclusion made specifically for the rumen). Yutin et al., 2015 characterized virophage genomes that are potentially parasitized on giant viruses (not in our scope). We want to point out that in this research article, the previous articles were not cited just because they are not less relevant than others.

We briefly mentioned the limitation of early rumen virome studies (lines 66-74). However, the major difference among reference 28, reference 29, and early rumen virome studies (Reference 18-20) is not in sequencing depth, but whether could identify viruses from the assembly de novo (in other words, whether or not novel viral genomes could be identified). Therefore, we highlighted these two studies.

We did not intend to criticize the studies for the small samples. We are aware that sample size can be limited if invasive procedures or animal welfare issues are involved, or when wild animals are sampled. We just wanted to discuss the results of those studies in the context of sample sizes, which are important. We also wanted to emphasize that viral genomic diversity could not be fully captured from the limited number of animals and limited feeding conditions.

Line 95, omit and it's implication... to... emissions, and change to rumen function. This paragraph also tries to explain the terminology used, vOTUs vs vMAGS, and ARGs, it doesn't read well – please revise.

AU: Revised. We removed the defined term vMAGs. For vOTUs and ARGs, they are commonly used in literature.

Additional note: Some of the Supplementary figures do not fit well on the page (I could not see all the figure legends).

AU: Resized.

Reviewer #2 (Remarks to the Author):

The revised manuscript "Unraveling the viral dark matter of the rumen microbiome with a new global virome database" by Yan et al. has incorporated additional analyses and experiments. However, the manuscript still lacks substantial innovation, has a limited research scope, and predominantly provides descriptive results that occasionally lack precision.

AU: This study was not intended to be a mechanistic study. Our main objective was to develop a large rumen virome database as a resource that can help future studies. We think this study did provide some new insights into the rumen virome in general, not specific to one breed or species fed a specific diet, such as the potential of some rumen viruses to contribute to cellulose digestion via their AMGs encoding cellulases, and infection of some species of the rumen microbiome by members of the core rumen virome.

Here are some major points:

1. One weakness of this study is the lack of sufficient data to substantiate the quality of the

proposed database. The analysis in Line113 is deemed inaccurate as the authors only utilize a subset of the datasets to validate the specificity of the database, which is derived from all the datasets. Consequently, the reported 10-fold expansion of rumen viruses may be overestimated. Additionally, the analysis in Line139 primarily focuses on host prediction, yet phage host prediction is a complex task that cannot be adequately resolved using a single phage database.

AU: As a rumen virome database, we think the most important quality is it contains no or few false vOTUs. We used the state-of-the-art benchmarked bioinformatics tool and conservative criteria to identify vOTUs to avoid or minimize false identification. We also checked all the vOTUs to make sure they meet the Uncultivated Virus Genome standards (MIUViG).

The analysis mentioned in line 113 was used to compare the new RVD database and the IMG/VR V3 database to see whether RVD improves the detection of viruses from the same 240 rumen metagenomes, not to validate the quality of the RVD quality. To make a fair comparison, we compared the two databases using another two sets of rumen metagenomes that have not been included in either database. The increase in the detection of viruses is based on that direct comparison. The new results are in line with the previous results. We want to show that RVD can improve virus detection in rumen metagenomes. We rephrased those sentences to clarify that.

We agree that phage host prediction is a challenging task. As mentioned above, we predicted the host with the most conserved methods and criteria and made sure the vOTUs meet the MIUViG standards. We also emphasized in the Discussion (line 358), the importance of experimental validation.

2. Line147-160: One aspect that needs improvement in this study is the clarity of the provided information. For instance, the statement "The authors could only assign the 2,243 viral contigs to families using BLASTn against NCBI RefSeq Viral" suggests that the cited study was able to assign a greater number of viruses compared to RVD. The writing is difficult to understand here. This part cannot be evaluated.

AU: What we meant was that the cited study could assign the 2,243 viral contigs only to families, not further down to genus or species. Because the cited study used Blastn for the assignment, which is too liberal and is now considered inappropriate, the comparison is not

appropriate. We deleted this part and emphasized that we could achieve genus-level taxonomy assignment, which could not be achieved by the original paper (lines 143-145). We edited the manuscript extensively, and we also have the manuscript edited by SpringerNature Author Services to improve clarity and readability.

3. Line 161: This does not seem the right way to compare VLP-enriched and shotgun metagenomic data. The big limitation of vOTU derived from metagenome is that, prophages, even defective ones that are not viruses, cannot be distinguished. This is the main cause of the results here. The data actually suggested real viral genomes have been overestimated in RVD.

AU: Phages are identified from both VLP-enriched and bulk metagenomes with several commonly used viral identification tools, such as VirSorter2, VIBRANT, CheckV, each of which has a set of criteria, including criteria requiring the presence of a minimum number of essential viral genes. However, they may not completely exclude defective or cryptic prophages that have lost one or more viral genes. Intuitively, more prophage sequences, including potential defective prophage sequences, might be found in bulk metagenomes than in VLP-enriched metagenomes, but one early study found more sequences of prophages than sequences of lytic phages in a VLP-enriched rumen metagenome (reference 18). In our study, bulk metagenomes did not recover a significantly higher proportion of lysogenetic phages (Supplementary Fig 3b).

RVD was developed from sequences recovered from bulk rumen metagenomes. The comparison between the VLP-enriched and the bulk rumen metagenome allowed us to evaluate if RVD can help identify viruses from VLP-enriched rumen metagenomes.

4. Line171-184: The writing is difficult to understand here. The term “Genus” and “Species” are very confusing. This part cannot be evaluated.

AU: We specified the “species” and “genera” we are referring to by adding “microbial” and “ciliates” to avoid confusion between viral and microbial species or genera. We also reworded this section for clarity.

5. Line192: “Lysogeny rate” is not a proper term, but extensively used in the manuscript. The method used to calculate “Lysogeny rate” is not clear.

AU: The term “lysogeny rate” is rephrased as “lysogeny rate (%)”. And it was used previously in Reference 5. The calculation of lysogeny rate was defined in M&M.

6. Line248: The new experiments did not contribute much to the manuscript. The authors are validating some known gene functions.

AU: The functions of GH5 and GH9 are known, but they are known to be present in some bacteria, fungi, and protozoa, not as viral AMGs. No study has identified any GH5 or GH9 as viral AMGs in any ecosystem. Actually, except for very few studies, virome studies only use bioinformatic analysis to identify AMGS, including the two rumen virome studies that reported AMGs (References 28 and 29). Considering the importance of cellulose digestion in the rumen, we experimentally verified if the GH5 and the GH9 AMGs do encode functional cellulases. We think the experimental verification is valuable and contributes to the understanding and appreciation of the impact that phages may have on their hosts.

Actually, experimental validation of AMG function has been listed as one of the outstanding questions in a recent review paper (Figure 1. Overview of the functional ecology of bacteriophages and current limitations. <https://doi.org/10.1016/j.mib.2022.102245>)

7. Line274: The reviewer expected a comprehensive analysis of virome in different ruminants. The “core” virome analysis did not contribute too much to the manuscript.

AU: As noted in the manuscript and responded to a comment of Reviewer 1, the virome profile in each animal is highly individualized. Rumen microbial profiles are primarily driven by the diets of animals, and thus rumen viral profiles will also be highly dependent on diets and animals. Hence, a comparison of rumen viral profiles should be done in the context of a specific diet and animals, but information on the diet was not reported for some of the rumen metagenomes we analyzed. Additionally, comparison results will need to be interpreted in the context of the microbiome. This goes beyond the scope and the objective of this study. Additionally, some animal species/breeds are represented by much more metagenomes than others (729 vs 82). This could lead to bias.

Since core virome has been reported for the human gut, we want to see if there is a core rumen virome, and if the members of the core rumen virome infect the members of the core rumen

microbiome. As we responded to one comment from reviewer 1 about the core virome, we repeated the analysis for the core virome but focused only on 283 cattle samples with reported dietary information. We grouped the 283 cattle samples based on the levels of dietary concentrate (low, medium, and high) and examined the vOTUs shared by samples within each concentrate level to determine the “core” virome at each concentrate level. Consistent with our previous analysis, the rumen virome profile is highly individualized. Considering the small sample size at each concentrate level, we did not emphasize the “core” for each concentrate level, but 14 vOTUs were found to be “core” across all three concentrate levels. 13 of them were predicted to infect *Prevotella* species. This suggests that at least some phages of the core rumen virome can infect the core rumen microbiome. The new analysis results are included in the revised manuscript.

8. The manuscript primarily presents descriptive results without a well-defined scientific narrative, and the innovative aspects of the bioinformatic methodology employed are limited.

AU: As responded above, this is not a hypothesis-driven mechanistic study. However, we think this study and the new RVD will be a useful resource for future research. Actually, the human gut virome is better studied and characterized owing to the several available large virus databases specific to the human gut ecosystem and viral-host linkage information. While previous rumen virus studies were hindered by a lack of a rumen virus database and viral-host linkage information. In addition to RVD, our results advance our understanding of the rumen virome, with respect to their diversity, the bacteria, archaea, and protozoa they may infect, and their potential impact on the rumen microbiome and functions.

We used state-of-the-art bioinformatics methodologies in our analyses. New bioinformatics tools are being developed, such as the very recent Phana developed for virus classification for human gut virome (<https://doi.org/10.1038/s41587-023-01799-4>). However, for such a new tool, or other tools, to be used for other ecosystems, niche-inclusive virome databases are still required. Thus, RVD will be useful in future studies even after new bioinformatics tools are available.

9. The writing is not in a proper way. Too many citations in the results, which made it difficult to read. The total number of references is almost 100, which is too much for a research article.

AU: We carefully revised and edited the text to improve the clarity and readability. SpringerNature Author Services also helped edit the manuscript to ensure clarity and readability. We delete those citations that are not absolutely needed.

Reviewer #3 (Remarks to the Author):

In this manuscript, the authors have provided substantially reorganized and revised version of their prior submission. While still focusing on the use of a globally-distributed collection of rumen metagenomes for the discovery of viruses, the authors have provided additional detail, context, and experimentally demonstrated activity of a viral auxiliary metabolic gene (cellulase). These updates represent a substantial improvement to the manuscript and would allow the work to be fully replicated by others.

AU: We appreciate the recognition and positive comments.

REVIEWERS' COMMENTS

Reviewer #1 (Remarks to the Author):

Thank you to the authors for addressing my previous review comments and I appreciate your thoughtful and polite responses. I feel I have given considerable time to the review of this paper and the third iteration is a considerable improvement on the first (and second) manuscripts. The language used is more appropriate and less awkward in places. Unfortunately, some errors have crept in as changes have been made, particularly in the introduction. I have addressed these in the comments below.

The primary focus of the manuscript is to detail a rumen virus database which the authors have prepared, using publicly available metagenomic datasets. This is now clearly described and as such, the manuscript is fine for publication. However, the manuscript also contains a considerable amount of interpretation, drawing information from the database and hypothesizing what this means. While the majority of this interpretation is fine now that the metadata behind the datasets used has been considered (relating the virome to the diet was a significant improvement), I have one major criticism of the current manuscript which prohibits me from accepting the manuscript as suitable for publication.

My major criticism of the current manuscript concerns the mention and interpretation of antibiotic resistance genes within the virome database. I mentioned that I did not think that this should be included when it first appeared in the second manuscript. Now the authors have taken this further, analysing the virome data to take into account geographical metadata i.e. the locations/countries from which the rumen metagenomes.

As I stated in the previous review, the use of antimicrobials in agriculture is a very contentious issue, with long-term public health implications. It also has geo-political implications for beef export markets. I do not think a publication whose primary aim is to describe a viral sequence database, should be overstating and potentially mis-interpreting their findings about the presence of antimicrobial genes, particularly in a journal as prestigious as Nature.

The data analysis described in the manuscript picked up relatively few antibiotic resistance genes, occurring at very low levels according to the data presented in Figure 5 – a maximum of 0.3 ARG carrying phages per billion bp (quite a confusing unit of gene coverage, but sounds like an extremely low incidence to me). The antibiotic resistance genes discovered may be naturally occurring (e.g. normal microbial efflux pumps) and convey resistance for antibiotics not even used in agriculture.

There is absolutely no way of validating this, as there is usually not any on-farm antibiotic usage data in the metadata available for the metagenomic datasets used in the study.

By bringing in the country of origin metadata, the authors, perhaps inadvertently, are essentially accusing the Chinese beef industry of poor antibiotic stewardship. This is a virome database paper, not a study designed to specifically investigate, compare and validate on-farm antimicrobial usage, and accurately assess the rise of antibiotic resistance in the beef and dairy industries of China, North America and the UK. The over-extrapolation in the results and discussion should not be included (remove the section from line 240 and the analysis using geographical metadata e.g. Fig 5b). While this is included in this manuscript, I think this manuscript is completely unsuitable for publication.

Additional comments:

Line 64, This sentence has been included following the second revision of this paper, it is incorrect. Prophages have NOT been identified from most of the sequenced rumen microbial genomes. The publication cited that mentioned rumen prophage sequences was a book chapter from 2015. Since then there have been hundreds of rumen bacteria genome sequenced (see the Seshadri et al 2018 Hungate 1000 genome sequencing paper), for which prophage sequences have not been included in publicly available databases.

Line 71 Clustered regularly interspaced short palindromic repeats (CRISPR), it is spelled correctly in other parts of the paper, but the first time this is mentioned (in the Introduction), this is misspelled – R missing. The first time an acronym is used it should be written in full – amend this please. Line 74 is very awkward, reword.

Again, we appreciate the valuable comments and suggestions. Please see below our responses. The changes in the main text are highlighted in blue, but the deletions are not shown.

REVIEWER COMMENTS

Reviewer #1 (Remarks to the Author):

Thank you to the authors for addressing my previous review comments and I appreciate your thoughtful and polite responses. I feel I have given considerable time to the review of this paper and the third iteration is a considerable improvement on the first (and second) manuscripts.

AU: Appreciate your recognition of the revision and improvement.

The language used is more appropriate and less awkward in places. Unfortunately, some errors have crept in as changes have been made, particularly in the introduction. I have addressed these in the comments below. The primary focus of the manuscript is to detail a rumen virus database which the authors have prepared, using publicly available metagenomic datasets. This is now clearly described and as such, the manuscript is fine for publication. However, the manuscript also contains a considerable amount of interpretation, drawing information from the database and hypothesizing what this means. While the majority of this interpretation is fine now that the metadata behind the datasets used has been considered (relating the virome to the diet was a significant improvement), I have one major criticism of the current manuscript which prohibits me from accepting the manuscript as suitable for publication.

My major criticism of the current manuscript concerns the mention and interpretation of antibiotic resistance genes within the virome database. I mentioned that I did not think that this should be included when it first appeared in the second manuscript. Now the authors have taken this further, analysing the virome data to take into account geographical metadata i.e. the locations/countries from which the rumen metagenomes.

As I stated in the previous review, the use of antimicrobials in agriculture is a very contentious issue, with long-term public health implications. It also has geo-political implications for beef export markets. I do not think a publication whose primary aim is to describe a viral sequence database, should be overstating and potentially mis-interpreting their findings about the presence of antimicrobial genes, particularly in a journal as prestigious as Nature.

The data analysis described in the manuscript picked up relatively few antibiotic resistance genes, occurring at very low levels according to the data presented in Figure 5 – a maximum of 0.3 ARG carrying phages per billion bp (quite a confusing unit of gene coverage, but sounds like an extremely low incidence to me). The antibiotic resistance genes discovered may be naturally occurring (e.g. normal microbial efflux pumps) and convey resistance for antibiotics not even used in agriculture. There is absolutely no way of validating this, as there is usually not any on-farm antibiotic usage data in the metadata available for the metagenomic datasets used in the study.

By bringing in the country of origin metadata, the authors, perhaps inadvertently, are essentially accusing the Chinese beef industry of poor antibiotic stewardship. This is a virome database paper, not a study designed to specifically investigate, compare and validate on-farm antimicrobial usage, and accurately assess the rise of antibiotic resistance in the beef and dairy industries of China, North America and the UK.

AU: We appreciate your constructive comments. We agree and we have deleted the ecological analysis related to ARG-encoding viruses. This includes the second half of the section regarding ARGs in Results, the first paragraph of the ecological analysis of Methods, and the panel of Fig. 5.

The over-extrapolation in the results and discussion should not be included (remove the section from line 240 and the analysis using geographical metadata e.g. Fig 5b). While this is included in this manuscript, I think this manuscript is completely unsuitable for publication.

AU: We have removed the statements pertaining to the potential risks of ARGs transmission through the food chain via rumen viruses. However, we retained the mention of ARG-encoding rumen viruses, as they detail the vMAGs identified in the RVD. Additionally, we emphasized the need for more research to validate the ecological importance of ARG-encoding viruses.

Additional comments:

Line 64, This sentence has been included following the second revision of this paper, it is incorrect. Prophages have NOT been identified from most of the sequenced rumen microbial genomes. The publication cited that mentioned rumen prophage sequences was a book chapter from 2015. Since then there have been hundreds of rumen bacteria genome sequenced (see the Seshadri et al 2018 Hungate 1000 genome sequencing paper), for which prophage sequences have not been included in publicly available databases.

AU: We deleted the inaccurate sentence.

Line 71 Clustered regularly interspaced short palindromic repeats (CRISPR), it is spelled correctly in other parts of the paper, but the first time this is mentioned (in the Introduction), this is misspelled – R missing. The first time an acronym is used it should be written in full – amend this please.

AU: Sorry for the oversight, the typo was corrected, and the full name was spelled out.

Line 74 is very awkward, reword.

AU: We rephrased the sentence.